# *GRIP*: A Graph-Based Reasoning Instruction Producer

**Jiankang Wang**[1][‡][*], **Jianjun Xu**[1][*], **Xiaorui Wang**[2], **Yuxin Wang**[1], **Mengting Xing**[2],
**Shancheng Fang**[2], **Hongtao Xie**[1][†]

[1]University of Science and Technology of China
[2]MetaStone Technology, Beijing, China
wangjiankang@mail.ustc.edu.cn

## Abstract

Large-scale, high-quality data is essential for advancing the reasoning capabilities of large language models (LLMs). As publicly available Internet data becomes increasingly scarce, synthetic data has emerged as a crucial research direction. However, existing data synthesis methods often suffer from limited scalability, insufficient sample diversity, and a tendency to overfit to seed data, which constrains their practical utility. In this paper, we present *GRIP*, a **G**raph-based **R**easoning **I**nstruction **P**roducer that efficiently synthesizes high-quality and diverse reasoning instructions. *GRIP* constructs a knowledge graph by extracting high-level concepts from seed data, and uniquely leverages both explicit and implicit relationships within the graph to drive large-scale and diverse instruction data synthesis, while employing open-source multi-model supervision to ensure data quality. We apply *GRIP* to the critical and challenging domain of mathematical reasoning. Starting from a seed set of 7.5K math reasoning samples, we construct **GRIP-MATH**, a dataset containing 2.1 million synthesized question-answer pairs. Compared to similar synthetic data methods, *GRIP* achieves greater scalability and diversity while also significantly reducing costs. On mathematical reasoning benchmarks, models trained with GRIP-MATH demonstrate substantial improvements over their base models and significantly outperform previous data synthesis methods.

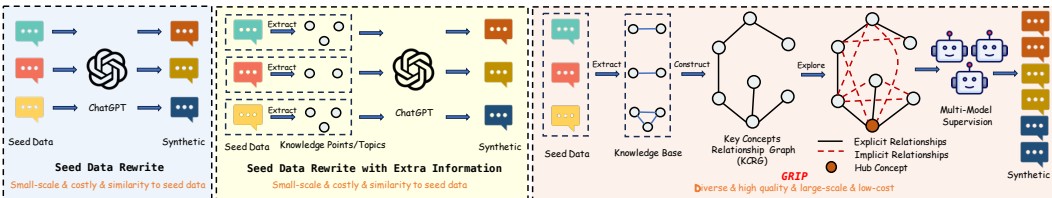

Figure 1: The two methods on the left reconstruct data based on either the seed data itself or extra information, making them similar to the seed and difficult to scale. Our method first extracts key concepts to construct a concept graph, then synthesizes novel questions by leveraging multiple relationships within the graph.

## 1 Introduction

In recent years, large language models (LLMs) have achieved remarkable performance across a wide range of linguistic tasks [1, 11, 30], largely driven by access to large-scale, high-quality training data. However, recent studies [23, 34]have shown that as the pool of high-quality publicly available data on the Internet continues to diminish, the further advancement of LLMs may be limited by data scarcity.

---

[‡]Work done during the internship at MetaStone Technology.
[*]Equal contribution.    [†] Corresponding author.

39th Conference on Neural Information Processing Systems (NeurIPS 2025).

As a result, the exploration of synthetic data has attracted increasing attention. Several leading commercial models [27, 38, 39] have incorporated large amounts of synthetic data as indispensable components of their training pipelines. Nevertheless, the details of these synthesis processes are often undisclosed, or their resource requirements are prohibitively high. Therefore, developing a practical and highly scalable method for synthesizing high-quality data has become a critical challenge for the continued evolution of large language models.

One approach for building high-quality reasoning datasets is data filtering [27, 40, 43]. This method involves extracting data from pre-training corpora such as Common Crawl and rewriting it using advanced commercial models or human annotation. However, due to these corpora's vast scale and inherent noise, data filtering results in high post-processing costs and inconsistent data quality and distribution. A more efficient approach is data synthesis [14, 19, 20, 21, 33, 41, 42]. Such approaches often leverage prompts to closed-source models [1, 30], rephrasing seed data or generating analogous questions to augment data, as shown in Figure 1. More advanced variants incorporate the knowledge points or topics of the seed data into the prompts, guiding the model to generate new questions centered around this extra information. However, these methods have several inherent limitations. First, since they only rephrase or slightly modify seed questions, the expansion in data volume is inherently limited, and does not achieve orders-of-magnitude growth. Second, the strong dependence on seed data means that the generated questions are often highly similar to the original seeds, restricting the diversity of the synthesized data. Third, the total amount of data that can be synthesized is further restricted by the high cost and limited accessibility of closed-source models, making large-scale generation impractical.

Inspired by human learning patterns, we move beyond directly manipulating seed data and instead focus on the underlying high-level concepts (such as topics and knowledge points) embedded within each example. Typically, a seed example consists of three or four key concepts. Prior methods, which rephrase questions or alter scenarios, in fact only rearrange the same set of core concepts, thus providing limited new information for the model to learn. To address this limitation, we first systematically extract all high-level concepts from the seed data and construct a co-occurrence graph, where two concepts are connected if they ever appear together in the same example. By designing new examples that combine concepts which have never co-occurred in the original data, we are able to synthesize truly novel data that lies outside the original seed distribution. We observe that the number of valid non-co-occurring concept pairs far surpasses co-occurring ones. Even a modest increase in seed data leads to a dramatic growth of novel concept combinations, highlighting the strong scalability enabled by this idea. Building on this idea, we propose **GRIP**: a graph-based framework to efficiently synthesize large-scale, high-quality and diverse reasoning data.

As presented in Figure 1 right, *GRIP* encompasses four steps: (1) Knowledge base construction: We begin by extracting key concepts (KCs) from seed data using a specialized model, such as mathematical theorems (e.g., the Pythagorean theorem), formulas (e.g., the quadratic formula), and key properties (e.g., the distributive law). This is followed by a filtering step to remove duplicates and low-quality key concepts. (2) Building the graph: The Key Concepts Relationship Graph (KCRG) is designed to capture the interconnections among concepts. In the graph, we define two types of relationships to form the KCRG: *explicit* and *implicit*. Explicit relationships are represented by solid lines, connecting concepts that originate from the same example. Implicit relationships, shown as dashed lines, connect concepts that are not directly related in the seed but exist within a certain distance of each other. These implicit relationships are further classified into two-hop and three-hop connections based on their distance. (3) Synthesis: The specialized model generates new samples by feeding its designed prompts and explicit and implicit concept combinations from the KCRG. (4) Data Evaluation: Multiple advanced open-source models are utilized to filter the synthesized data by jointly scoring the new samples.

To demonstrate the effectiveness of GRIP, we apply it to one of the most challenging reasoning domains: mathematics. We use 7.5K question-solution pairs from the MATH training set as seed data and synthesize a new dataset, **GRIP-MATH**, which contains over 2.1 million math question-solution pairs. We use GRIP-MATH to train large language models (LLMs) with diverse architectures and parameter sizes, including Qwen1.5-7B[3], Mistral-7B [17], LLaMA3-8B[22], LLaMA3.1-8B [10], Qwen2-1.5B[37], and Qwen2-7B [37]. On mathematical reasoning benchmarks, models trained with GRIP-MATH demonstrate substantial improvements over their base models and significantly outperform previous data synthesis methods. Furthermore, GRIP-MATH also enhances scientific reasoning performance, highlighting the strong generalization ability of GRIP.

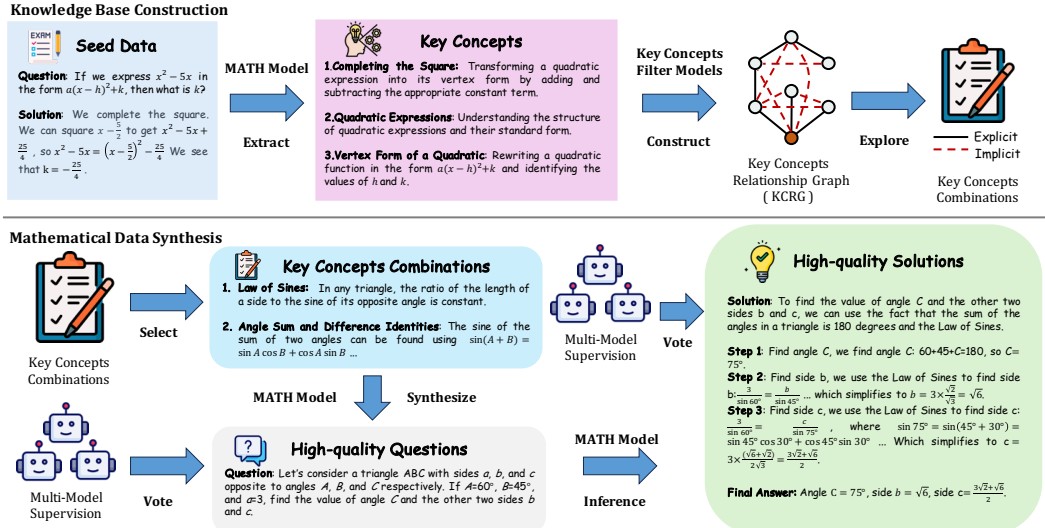

Figure 2: The overview of the Graph-based Reasoning Instruction Producer (GRIP). GRIP begins with seed data and follows a four-step process: (1) knowledge base construction, (2) key concepts relationship graph construction, (3) graph-based synthesis, and (4) evaluation by multiple models voting. After these steps, we obtain the GRIP-MATH dataset.

## 2 Related work

### 2.1 LLMs and Mathematical Reasoning

Recent years have seen remarkable advances in large language models (LLMs) for mathematical reasoning. Various strategies have been proposed to further improve LLMs' math abilities, including chain-of-thought prompting [36], program synthesis [27, 35], and fine-tuning on high-quality math corpora [21]. More recently, proprietary models trained on extensive curated datasets and large-scale synthetic data—such as DeepSeekMath [27] and Qwen2.5-Math [39]—have achieved state-of-the-art results. However, the lack of transparency in their data synthesis methods limit reproducibility. In response, research has increasingly focused on synthetic data generation [29, 43] as a scalable and reproducible approach to b oosting LLM performance on mathematical reasoning tasks.

### 2.2 Data Synthesis

With the imminent exhaustion of Internet data, data synthesis has drawn increasing research attention. In mathematical reasoning, data synthesis is primarily used for instruction fine-tuning, where each sample consists of a question text and its corresponding answer. Our method can synthesize large-scale, high-quality mathematical inference data from limited seed data, making it suitable for continued pre-training tasks. Research efforts primarily concentrate on two pivotal aspects: improving data quality and generating novel questions. Regarding the generation of novel questions, one approach [19, 29, 33, 41, 42] entails rewriting or generating similar questions based on seed data for data augmentation. Another approach [14, 15, 20] involves generating new questions using knowledge points, either by generating new knowledge points via GPT-4 or extracting them from existing knowledge point databases. However, these approaches often suffer from limited scalability, high cost, and high similarity to seed data, due to their reliance on explicit relationships and closed-source models. Our method addresses these limitations by exploring both explicit and implicit relationships between key concepts using the graph and leveraging open-source models for cost-effective data synthesis.

## 3 Proposed Method

As presented in Figure 2, this section introduces a Graph-based Reasoning Instruction Producer (GRIP), a unified synthetic data framework with four steps: Knowledge Base Construction, Key Concepts Relationship Graph Construction, Synthesis Based on Diverse Key Concept Combinations,

Multi-Model Evaluation, and Dataset Statistics. Detailed descriptions and implementation steps for each component are provided in the subsequent sections.

## 3.1 Knowledge Base Construction

To enable large language models (LLMs) to more effectively process complex information such as mathematical problems, we first decompose each seed instance into its essential conceptual components, constructing a knowledge base that provides a structured representation of the overall dataset. In principle, a problem may be characterized by multiple hierarchical attributes, such as "Subject" (e.g., "Mathematics"), "Topic" (e.g., "Algebra"), and "Key Concept" (e.g., "Permutations and combinations" or "Difference of cosines formula"). For simplicity and to facilitate both concept extraction and automated modeling, we focus exclusively on extracting "Key Concept" as the key conceptual features for each question. Empirically, we find that this abstraction is sufficient to capture the salient mathematical characteristics of most problems.

GRIP uses the MATH training set as the seed data which consists of 7.5k math problems. We first extract no more than 5 relevant key concepts (KCs) from each seed problem with prompt engineering of Qwen2.5-32B [38] (refer to Prompt A.1 in Appendix A.1). After extracting the KCs, we employ an embedding model [4] and Qwen2.5-7B [38] for dual filtering of the KCs. Initially, the LLM filters out KCs with vagueness, math errors, or excessive details. Subsequently, the embedding model clusters KCs with similar meanings, followed by a second validation using the LLM. Finally, the most appropriate and accurate KC from each cluster is chosen to represent the cluster, resulting in 10K qualified key concepts. More details about dual filtering and KC examples can be found in Appendix D.

## 3.2 KCRG Construction

To organize the disordered KCs in the knowledge base and explore their specific interconnections, we designed a Key Concepts Relationship Graph (KCRG) to capture the associations between KC pairs.

In the KCRG, each node is represented as a KC, and each solid edge represents that the connected KCs have co-occurred in the same problem. Specifically, the KCRG $\mathbb{G}$ can be represented as $\mathbb{G} = (\mathbb{K}, \mathbb{E})$. The nodes $\mathbb{K}$, which refer to key concepts, are denoted as $\mathbb{K} = \{\mathbf{k}_1, \mathbf{k}_2, \ldots, \mathbf{k}_{|\mathbb{K}|}\}$. The edges $\mathbb{E}$ are denoted as $\mathbb{E} = \{\mathbb{E}_{\text{ex}}, \mathbb{E}_{\text{im}}\}$, and there are two types: (1) Explicit ($\mathbb{E}_{\text{ex}}$): Key concepts that appear together in the same seed problem are connected by a solid edge. The explicit edges $\mathbb{E}_{\text{ex}}$ can be denoted as $\mathbb{E}_{\text{ex}} = \{(\mathbf{k}_i, \mathbf{k}_j) | \mathbf{D}(\mathbf{k}_i, \mathbf{k}_j) = 1\}$, where the edge distance $\mathbf{D}(\mathbf{k}_i, \mathbf{k}_j)$ represents the number of solid edges in the shortest path between $\mathbf{k}_i$ and $\mathbf{k}_j$. Additionally, the edge weight $\mathbf{W}(\mathbf{k}_i, \mathbf{k}_j)$ is recorded to denote the co-occurrence frequency between $\mathbf{k}_i$ and $\mathbf{k}_j$. (2) Implicit ($\mathbb{E}_{\text{im}}$): Key concepts with more than one solid edge between them are connected by a dashed edge, as visualized in Figure 3. The implicit edges $\mathbb{E}_{\text{im}}$ can be denoted as $\mathbb{E}_{\text{im}} = \{(\mathbf{k}_i, \mathbf{k}_j) | \mathbf{D}(\mathbf{k}_i, \mathbf{k}_j) > 1\}$.

## 3.3 Synthesis Based on Diverse Key Concept Combinations

Different from previous methods that primarily relied on seed data and explicit relationships between KCs, we fully integrate both explicit and implicit relationships to generate more diverse synthetic data. We observed that incorporating overly distant implicit relationships tends to increase the proportion of low-quality data. This is because such KCs exhibit weaker correlations, making it difficult to synthesize meaningful questions. Therefore, we proposed four types of key concept relationships: one-hop, two-hop, three-hop, and community. Exploring implicit relationships has mined more key concept combinations, which is a key driver of GRIP's high scalability. Figure 3 illustrates an example of KCRG construction.

**One-hop** relationships represent explicit links between pairs of key concepts (KCs) that are directly connected by a single edge in the graph. The edge weight reflects the frequency of their co-occurrence in the seed data. Since these combinations are already present in the seed set, they exhibit high semantic relevance.

**Two-hop** relationships capture implicit connections, involving pairs of KCs that are connected indirectly through an intermediate node—i.e., two KCs that share a common neighbor. This indirect association often reflects shared characteristics or themes, thus preserving a moderate level of semantic relevance.

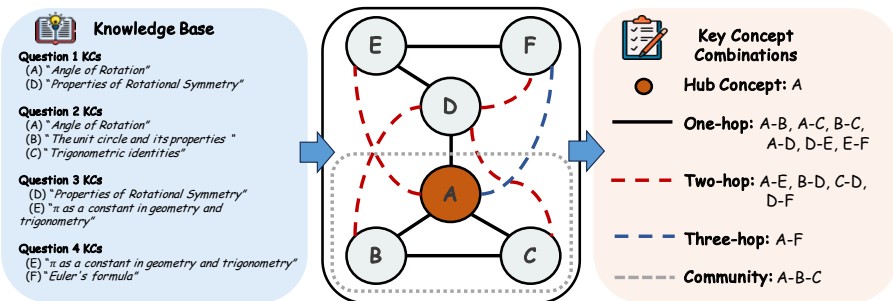

Figure 3: An example of constructing the Key Concepts Relationship Graph (KCRG) from an existing knowledge base and identifying the four key concept combinations we proposed.

**Three-hop** further explores implicit relationships by considering pairs of key concepts (KCs) that are three edges apart in the graph. In this setting, we focus on hub concepts —those nodes in the knowledge concept relationship graph (KCRG) that have the largest number of connections (i.e., high degree). Such hub concepts serve as central points in the knowledge network, indicating their broad relevance and significance across various topics. A three-hop combination is defined as a pair consisting of a hub concept and another KC that is three edges away from it. As the graph grows larger, multiple hub concepts may exist. Since the semantic relevance between KCs tends to decrease as the path length increases, we restrict three-hop relationships to only those involving hub concepts to help maintain meaningfulness. Additionally, we filter out low-weight three-hop combinations to further ensure that the selected three-hop pairs are relevant and informative.

For example, assume that a hub concept is "calculus". In the seed data, problems only associate "calculus" with fundamental concepts such as "limits" and "derivatives". However, in the KCRG, "calculus" is likely not adjacent but close to KCs such as "Fourier transforms" and "complex functions". By integrating these three-hop KCs (the same applies to two-hop) to construct novel problems, we increase the diversity of the problem set.

**Community** represents explicit relationships involving three or four key concepts (KCs), where every pair within the group is mutually connected by edges, forming a fully connected subgraph. Such communities indicate a strong correlation among the KCs and typically represent cohesive knowledge areas.

Accordingly, one-hop combinations are used to synthesize high-quality variant problems directly related to the seed data. Implicit relationship combinations are used to synthesize new distribution data, increasing the diversity of the dataset. Community-based combinations are used to synthesize integrative problems that require simultaneously applying multiple closely related key concepts.

As shown in Figure 3, after extracting the KCs from the seed, we construct the KCRG based on their co-occurrence. All key concept combinations in the graph that meet these four relationship types are extracted, and low-weight implicit relations are filtered out. We input the prompt and key concept combinations into the Qwen2.5-32B [38] to synthesize new problems. Different from other methods, we do not include seed data in the prompt, as this would cause the model to generate problems too similar to them. Before solution generation, a rating model assigns a difficulty level to each problem. For medium and low-difficulty issues, Qwen2.5-Math-7B [39] generates the solutions, while Qwen2.5-Math-72B handles high-difficulty problems. The complete prompt template is provided in Appendix A.2, Prompt A.2.

### 3.4 Multi-Model Evaluation

To match the evaluation effectiveness of closed-source models, we employ a multi-model supervision framework using three state-of-the-art open-source mathematical LLMs: DeepSeek-R1-Distill-Qwen-7B [11], Qwen2.5-Math-Instruct-7B [39], and DeepSeek-Math-RL [27]. These models are jointly used to score and filter the synthesized data, ensuring high quality of both problems and solutions.

For problem evaluation, we use a weighted scoring filtering strategy. Problems are evaluated on two criteria: logical completeness (absence of mathematical errors and accurate relation to provided key concepts) and presentational completeness (clarity, thoroughness, and absence of prompts or answers). Each model assigns a score between 0 and 1 for every problem. A weighted average

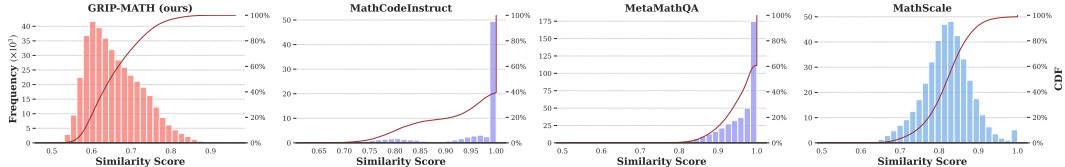

Figure 4: Histogram of the similarity scores between synthesized data and the seed data, including a comparison of GRIP-MATH with three open-source datasets in terms of seed similarity. The bars represent the frequency of the similarity scores, while the red line represents the cumulative distribution function (CDF) of the scores. It can be observed that the similarity scores of GRIP-MATH are concentrated between 0.55 and 0.65, whereas those of the other three methods are more concentrated around 0.85 or even 1. This indicates that GRIP-MATH exhibits lower seed similarity.

Table 1: Comparison of various methods in expansion ratio ($\times$) and synthesis cost ($10^{-2}$ cents). The expansion ratio represents the proportion of synthesized data to seed data. Synthesis Cost indicates the expenses associated with closed-source models or GPU usage for synthesizing a single data sample. Details of cost calculation are provided in Appendix B.

| Method | Data Source | Synthesis Model | Total Seed Data | Total Synthesized Data | Expansion Ratio | Cost | Novelty Rate |
|---|---|---|---|---|---|---|---|
| MetaMath [41] | GSM8K+MATH | GPT-3.5 | 15K | 395K | 26 | 23 | 17.9 |
| MathScale [29] | MWPBENCH | GPT-3.5 | 20K | 2M | 100 | 23 | 37.5 |
| WizardMath [21] | GSM8K+MATH | GPT-4 | 15K | 96K | 6.4 | 220 | - |
| XwinMath [19] | GSM8K+MATH | GPT-4 | 15k | 1.4M | 93 | 220 | - |
| MAmmoTH [42] | MAmmoTH datasets | GPT-4 | 220K | 262K | 1.2 | 220 | - |
| MathCoder [35] | GSM8K+MATH | GPT-4 | 15K | 80K | 5.3 | 220 | 9.1 |
| **GRIP** | MATH | Open-Source Model | 7.5K | **2.1**M | **280** | **0.57** | 71.8 |

score is then computed, where weights are proportional to each model's demonstrated mathematical ability. Problems with scores below 0.85 are discarded. For solution evaluation, we implement a strict single-vote veto mechanism. A solution must be mathematically correct and fully address all aspects of the problem. The solution is scored as 0 or 1 by each model, and only those unanimously approved by all models (perfect score) are retained. Any solution receiving a negative vote from any model is filtered out to maintain overall dataset quality. After evaluation of problems and solutions, we constructed a dataset named GRIP-MATH comprising 2.1 million high-quality mathematical questions at low cost.

## 3.5 Dataset Statistics

To demonstrate the superiority of our methodology, we compare various data synthesis methods from multiple dimensions. (1) For **scalability**, we compare the seed and synthesized data volumes among different methods. As illustrated in Table 1, MathScale [29] is the largest dataset, utilizing GPT-3.5 to expand 20K seed data into 2M samples, reaching a 100-fold expansion ratio. Methods employing GPT-4 for data synthesis demonstrate relatively low expansion ratios due to their high operational costs. In contrast, our approach expands 7.5K seed data to 2.1M samples, reaching the highest expansion ratio of 280-fold. (2) For **synthesis cost**, we calculate the per-sample synthesis cost for each method. To simplify the comparison, we consider API usage[*] costs for methods relying on commercial models and GPU computational costs[†] for our approach. As shown in Table 1, our method's per-sample synthesis cost is merely 2% of GPT-3.5-based methods and less than 1% of GPT-4-based methods. (3) For **data quality**, we compare the similarity between open-source datasets and their corresponding seed data using an embedding model. Specifically, we calculate the similarity between the embedded synthesized data and seed data to obtain the similarity score distribution for each dataset, as visualized in Figure 4. The results show that the rewritten methods of MathCoder [35] and MetaMath [41] show extremely high similarity to the seed data. Although MathScale is not directly based on rewriting, it still exhibits relatively high similarity due to its reliance on explicit relationships in the seed data. In comparison, GRIP-MATH has over 50% of its data similarity below 0.65, with the majority under 0.75 and none exceeding 0.9. (4) For **diversity**, we further analyzed the diversity of generated datasets by calculating the proportion of questions whose key concept combinations were not present in the seed data. This metric reflects how many genuinely novel questions each method can generate. We found that for other methods, this proportion was consistently below 40%, indicating that they generated relatively few truly novel questions. In contrast, thanks to GRIP's use of implicit relationships, we generated 1.5 million novel questions,

---

[*]https://openai.com/api/pricing/

[†]https://power.netmind.ai/rentIntro

Table 2: The performance of models on mathematical reasoning tasks. The results are sourced from the evaluation scripts of MAmmoTH2 and OpenCompass. GK II denotes the 2010-2022 Math II MCQs from GAOKAO-Eval, and GK I represents the 2010-2022 Math I MCQs. ∗ denotes our reproduced results based on the officially released codes.

| Model | Base | Size | MATH | GSM8K | GK II | GK I | SVAMP | AVG |
|---|---|---|---|---|---|---|---|---|
| *Specific Models* | | | | | | | | |
| Qwen2-Math | Qwen2 | 1.5B | 44.4 | 71.3 | 57.3 | 50.0 | 76.4 | 59.9 |
| Qwen2-Math | Qwen2 | 7B | 50.4 | 81.2 | 78.9 | 62.5 | 88.1 | 72.2 |
| DeepseekMath-Instruct | DeepseekMath | 7B | 46.8 | 82.9 | 58.3 | 46.7 | 84.0 | 63.7 |
| DeepseekMath-RL | DeepseekMath | 7B | 51.7 | 88.2 | 61.5 | 58.9 | 86.4 | 69.3 |
| *Base Models* | | | | | | | | |
| Mistral-7B | - | 7B | 11.2 | 36.2 | 13.8 | 12.2 | 66.9 | 28.0 |
| Qwen2 | - | 1.5B | 21.7 | 58.5 | 29.8 | 28.5 | 67.4 | 41.2 |
| Qwen2 | - | 7B | 45.2 | 80.3 | 66.5 | 52.8 | 87.5 | 66.5 |
| Qwen1.5 | - | 7B | 13.3 | 54.1 | 56.4 | 53.7 | 73.4 | 50.2 |
| LLaMA3 | - | 8B | 21.3 | 54.8 | 4.1 | 7.9 | 69.7 | 31.6 |
| LLaMA3.1 | - | 8B | 23.1 | 54.9 | 10.6 | 10.8 | 70.1 | 33.9 |
| *Data synthesis method* | | | | | | | | |
| MetaMath | Mistral | 7B | 28.2 | 77.7 | 9.2 | 9.4 | 77.2 | 40.3 |
| WizardMath | Mistral | 7B | 31.0 | 78.0 | 17.0 | 15.4 | 48.5 | 38.0 |
| MathCoder-CL | Mistral | 7B | 30.2 | 67.8 | 9.6 | 15.9 | 70.7 | 38.8 |
| MathScale | Mistral | 7B | 34.5 | 74.0 | 36.7 | 31.3 | 79.6 | 51.2 |
| MathScale* | Qwen1.5 | 7B | 32.2 | 69.6 | 55.2 | 52.4 | 75.1 | 56.9 |
| MAmmoTH | Mistral | 7B | 18.2 | 61.5 | 22.0 | 21.5 | 71.7 | 39.0 |
| MAmmoTH2 | Mistral | 7B | 36.7 | 68.4 | 44.9 | 29.4 | 81.8 | 52.2 |
| MAmmoTH2 | LLaMA3 | 8B | 35.8 | 70.4 | 33.5 | 24.3 | 78.6 | 48.5 |
| *GRIP Model Trained only with GRIP-MATH* | | | | | | | | |
| GRIP | Mistral | 7B | 41.6 | 83.5 | 42.9 | 33.1 | 85.4 | 57.3 |
| GRIP | Qwen1.5 | 7B | 37.9 | 77.1 | 57.4 | **56.3** | 80.8 | 61.9 |
| GRIP | LLaMA3 | 8B | 37.2 | 76.5 | 38.5 | 31.8 | 82.2 | 53.2 |
| GRIP | LLaMA3.1 | 8B | 37.1 | 72.0 | 44.5 | 35.1 | 84.2 | 54.6 |
| GRIP | Qwen2 | 1.5B | 41.1 | 74.9 | 51.6 | 44.3 | 80.9 | 58.6 |
| GRIP | Qwen2 | 7B | **53.4** | **86.0** | **68.4** | 54.2 | **88.8** | **70.2** |

with 71.8% of key concept combinations not found in the seed set. This demonstrates a substantial improvement in diversity achieved by our approach.

# 4 Experiments

## 4.1 Training Setup

We selected Qwen1.5-7B [3], Mistral-7B [17], LLaMA3-8B [22], LLaMA3.1-8B [10], Qwen2-1.5B [37], and Qwen2-7B [37] as baseline models, and trained all of them exclusively on the GRIP-MATH dataset. The fine-tuning is performed using the LLaMAFactory [44] framework over 2 epochs, with a learning rate of 5e-6, a global batch size of 128, and a maximum sequence length of 4096. A cosine schedule with a 3% warm-up ratio is adopted to regulate the learning rate. For expedited and efficient training, we leveraged DeepSpeed [25] ZeRO Stage 3 and FlashAttention 2 [9]. The synthesis with GRIP was completed in 36 hours using 8 NVIDIA A100 GPUs and vLLM [18].

## 4.2 Evaluation Datasets

To rigorously assess the enhancement in mathematical reasoning capabilities of models trained with GRIP-MATH, we employed a suite of mathematical evaluation datasets, including GSM8K [7], MATH [13], GAOKAO-Eval [31] and SVAMP [24]. In addition, to assess the impact of GRIP-MATH on the model's reasoning capabilities across other domains (e.g., physics, chemistry, coding, and logic), we evaluate the model using a range of scientific reasoning datasets, including ARC-C [6], MMLU-STEM [13], GPQA-Diamond [26], BBH [28], TheoremQA [5], and MBPP [2]. The results are sourced from the evaluation scripts of MAmmoTH2 [43] and OpenCompass [8].

Table 3: Results on scientific reasoning tasks.

| Model | Base | Size | ARC-C | MMLU-STEM | GPQA-Diamond | BBH | TheoremQA | MBPP | AVG |
|-------|------|------|-------|-----------|--------------|-----|-----------|------|-----|
| Mistral | - | 7B | 74.2 | 50.1 | 24.7 | 55.7 | 19.2 | 47.5 | 45.2 |
| Qwen1.5 | - | 7B | 75.6 | 45.5 | 26.7 | 45.2 | 14.2 | 52.1 | 43.2 |
| LLaMA3 | - | 8B | 78.6 | 55.6 | 27.2 | 61.1 | 20.1 | 54.9 | 49.6 |
| LLaMA3.1 | - | 8B | 79.5 | 54.7 | 24.2 | 62.8 | 20.9 | 57.2 | 49.9 |
| Qwen2 | - | 1.5B | 60.5 | 42.9 | 23.7 | 36.8 | 15.1 | 36.9 | 36.0 |
| Qwen2 | - | 7B | 83.6 | 64.3 | 32.3 | 61.7 | 33.5 | 60.7 | 56.0 |
| GRIP | Mistral | 7B | 78.6 | 58.6 | 31.7 | 61.1 | 26.5 | 54.9 | 51.9 |
| GRIP | Qwen1.5 | 7B | 77.2 | 56.9 | 30.0 | 51.2 | 22.4 | 53.7 | 48.6 |
| GRIP | LLaMA3 | 8B | 80.5 | 60.8 | 30.8 | **63.7** | 24.2 | 58.4 | 53.1 |
| GRIP | LLaMA3.1 | 8B | 82.7 | 61.8 | 32.8 | 63.2 | 25.9 | 59.2 | 54.3 |
| GRIP | Qwen2 | 1.5B | 61.0 | 43.1 | 25.9 | 35.2 | 18.2 | 37.8 | 36.9 |
| GRIP | Qwen2 | 7B | **84.3** | **65.9** | **33.4** | 62.5 | **34.8** | **65.9** | **57.7** |

## 4.3 Main Results

Table 2 summarizes the performance of various models on a suite of mathematical reasoning benchmarks. Our experimental findings highlight three main observations:

**Substantial improvements over base models.** All GRIP-trained models significantly outperform their respective base models across all benchmarks. For example, Mistral-7B, when trained only on GRIP-MATH, achieves an average score of 57.3, compared to 28.0 for the vanilla Mistral-7B—a nearly 30-point improvement. Similar trends are observed for Qwen2 and LLaMA3 families, indicating the large and robust gains brought by high-quality GRIP-MATH data regardless of base model architecture or parameter size.

**Outperforms previous data synthesis methods by a large margin.** Compared to other open-source data synthesis methods such as MetaMath [41], WizardMath [21], MathScale [29], and MAmmoTH2 [43], our GRIP-trained models consistently obtain much higher scores. For instance, Mistral-7B with GRIP-MATH achieves 57.3 average, dramatically surpassing the best prior method (MathScale, 51.2). On the MATH benchmark, GRIP-7B reaches 41.6, far beyond the scores of WizardMath or MetaMath (31.0 and 28.2, respectively), confirming the effectiveness of our approach in generating high-quality, diverse mathematical reasoning data.

**Competitive with proprietary math specialist models.** Remarkably, GRIP-trained models close much of the gap between open-source foundation models and proprietary specialist models that leverage substantially more diverse and larger-scale training data (including web data, textbook corpora, exam problems, and Chinese data). On key benchmarks such as MATH, GSM8K, and especially on challenging out-of-domain datasets (GK II, SVAMP), GRIP models achieve performance on par with, or even surpass, proprietary models like DeepSeekMath-RL [27] and Qwen2-Math-7B [39], despite relying solely on GRIP-MATH synthetic data.

Overall, our findings show that GRIP-MATH brings substantial gains to base models, consistently outperforms prior data synthesis methods, and enables competitive or even superior performance to commercial specialist models on major benchmarks, all without using proprietary resources.

## 4.4 Results on Scientific Reasoning Benchmark

GRIP-MATH is constructed solely from the training set of MATH and does not incorporate any data from other datasets. Nevertheless, experiments show that GRIP-MATH not only enhances the model's mathematical reasoning abilities, but also brings significant improvements on out-of-domain scientific reasoning tasks. As shown in Table 3, we evaluate our models on several widely used datasets covering physics, biology, chemistry, and computer science. Across all benchmarks, GRIP-trained models consistently outperform their respective base models, demonstrating strong cross-domain generalization. For example, GRIP-trained Qwen2-7B achieves an average score of 57.7, compared to 56.0 for the base model, with notable gains on MMLU-STEM, GPQA-Diamond, and MBPP. These results highlight the strong generalization ability of GRIP-MATH across scientific domains, even without explicit exposure to additional out-of-domain training data.

## 4.5 Ablation Studies about GRIP

**Comparison between Multi-Model and GPT-4.1.** To investigate the difference between multi-model quality evaluation and using GPT-4.1 alone, we conducted a manual annotation of 500 synthesized math problems. Each sample was independently labeled as "qualified" or "unqualified" by human annotators. We then used various combinations of open-source models (DeepSeek-R1-Distill-Qwen-7B [11], Qwen2.5-Math-Instruct-7B [39], and DeepSeek-Math-RL [27]) and GPT-4.1 to score the same data: a problem was considered qualified if its question score was above 0.85 and its solution score was 1. Finally, we compared the model evaluation results with the manual labels, as shown in Table 5. Table 5 shows that three open-source models outperform GPT-4.1 in accuracy, demonstrating that multi-model supervision is an effective and economical substitute for closed-source evaluation.

Table 4: Ablation on Hop Distance

| Method | AVG Score |
|---|---|
| One-hop | 0.94 |
| Two-hop | 0.72 |
| Three-hop | 0.62 |
| Three-hop w/o hub KCs | 0.39 |
| Four-hop w/o hub KCs | 0.21 |

Table 5: Ablation on Model Combinations

| Model Combination | ACC |
|---|---|
| GPT-4.1 | 94.3 |
| Qwen2.5-Math-Instruct-7B | 81.4 |
| Qwen2.5-Math-Instruct-7B, DeepseekR1-7B | 90.5 |
| Qwen2.5-Math-Instruct-7B, DeepseekR1-7B, DeepseekMath-RL | **95.7** |

**Key Concept Filtering.** The quality of key concepts is crucial for subsequent data synthesis. We synthesize questions (excluding solutions) using both unfiltered and filtered key concepts, and evaluate them using the multi-model scoring system to calculate the average score for each setting. As shown in Table 6, applying key concept filtering increases the average question score by 0.2 points. This improvement is due to the removal of meaningless or incorrect key concepts, which would otherwise mislead the model and lead to lower-quality questions.

Table 6: Ablation on Filtering

| Method | AVG Score |
|---|---|
| w Concept Filtering | 0.83 |
| w/o Concept Filtering | 0.61 |

**The quality of data synthesized from different hop distances.** We compared the average scores of data synthesized from one-hop, two-hop, three-hop, and more distant. The results in Table 4 show that as the distance between key concepts increases, the average score of the synthesized data decreases. This is because meaningful combinations are rarer between more distant concepts, leading to a drop in data quality. Nevertheless, it is still possible to synthesize high-quality data from these distant combinations, although this typically comes at a higher synthesis cost. Moreover, utilizing hub concepts to facilitate high-hop data synthesis is an effective strategy for maintaining quality.

**Impact of Implicit and Explicit Data on Model Performance.** To investigate the effects of implicit and explicit data on model performance, we conducted a comprehensive ablation study. We first randomly sampled 0.2 million examples from the one-hop (explicit) dataset and, based on the same key concept combina-

Table 7: Ablation on Datasets

| Datasets | Math | GSM8K |
|---|---|---|
| one-hop, one-hop(duplication) | 29.3 | 67.6 |
| one-hop, two-hop, three-hop | 32.6 | 71.2 |

tions, synthesized another 0.2 million one-hop samples. Additionally, we randomly selected a total of 0.2 million samples from the two-hop and three-hop (implicit) datasets. We then trained Mistral-7B separately with the explicit data group and the implicit data group. As shown in Table 7, the results suggest that introducing novel types of questions is more beneficial for model training than continuing to expose the model to repeated samples of the same type.

## 5 Conclusion and Future Work

In this paper, we introduce GRIP, an efficient paradigm for synthesizing high-quality data. Utilizing this method, we construct the GRIP-MATH dataset, comprising 2.1 million high-quality question-solution pairs. By leveraging this dataset, GRIP models have demonstrated outstanding performance across mathematical and scientific reasoning benchmarks. Our research indicates that thoroughly exploring implicit knowledge relationships enables larger-scale and more diverse data synthesis; additionally, multi-model evaluation can approach closed-source performance while maintaining cost-effectiveness. Intuitively, GRIP should be applicable to various domains where data can be decomposed into key concepts; however, our current experiments are limited to the mathematics domain, and its effectiveness in other areas is yet to be verified, as we discuss in detail in Appendix F.

## Acknowledgment

This work is supported by the National Nature Science Foundation of China (62425114, 62121002, U23B2028, 62232006). We thank MetaStone Technology for providing closed-source model APIs support and GPU computing resources. We acknowledge the support of the GPU cluster built by MCC Lab of Information Science and Technology Institution, USTC, and USTC super-computing center for providing computational resources for this project.

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

# A Prompts

**Prompt A.1 : Prompt for Key Concepts Extraction**

You will be given a mathematics problem and its detailed solution. Please extract 1 to 5 key concepts according to the strict requirements below:

1. **Use precise mathematical terminology.** Each key concept must be named using accurate and professional mathematical terms; avoid colloquial, vague, or general language.

2. **Direct and exclusive relevance.** Only include key concepts that are directly applied or explicitly referenced in both the problem and its solution. Exclude unrelated, unused, or general concepts, even if they appear incidentally.

3. **Be as specific as possible.** Do not use broad descriptions like "basic algebra" or "elementary mathematics." Specify particular theorems, formulas, properties, or standard techniques (e.g., "Pythagorean theorem," "Closure property of multiples under addition").

4. **No repetition or composite items.** Each key concept must correspond to a single, unique, and atomic mathematical concept. Do not combine multiple theorems, laws, or properties into one item; do not list overlapping or paraphrased concepts.

5. **Do not include procedural descriptions or general skills.** Only extract concrete mathematical facts, such as theorems, definitions, formulas, or standard properties. Do not include step-by-step methods or broad problem-solving strategies (e.g., "expand the equation," "calculate the sum").

---

**Prompt A.2 : Prompt for New Problem Generation**

Please design a brand-new mathematics problem that integrates both "[Key Concept A]" and "[Key Concept B]", according to the following requirements:

1. The problem must organically combine the two key concepts within a single mathematical scenario or real-life context. Do not simply split them into two independent sub-questions.

2. The content of the problem should be free of logical and mathematical errors or ambiguities; the conditions must be sufficient and clearly stated.

3. The problem should have a certain level of difficulty, providing a challenge and thoroughly testing students' ability to apply knowledge comprehensively.

4. The context can be realistic or innovative, but the question must remain coherent and natural. You may introduce additional key concepts as needed to create a well-formed and meaningful problem.

5. Expression should be concise and clearly structured, without unnecessary or irrelevant information.

6. The problem should have a unique and definite answer or solution path.

7. Choose an appropriate question type (such as free response, proof, fill-in-the-blank, or multiple choice) according to the chosen key concepts.

# B Calculation of Synthesis Cost

For synthesis cost, we posit that methods using closed-source models incur cost solely from the closed-source model cost[‡]; whereas for our method, we only need account for GPU usage cost[§]. Based on information from the web pages, the input cost of GPT-4 is \$10 per 1M tokens, and the output cost is \$30 per 1M tokens. For GPT-3.5, the input cost is \$1.5 per 1M tokens and the output cost is \$2 per 1M tokens. The cost of using one NVIDIA RTX A100 (80G) is \$0.42 per hour.

According to our experiments, synthesizing and scoring problems and solutions requires at least 1000 input tokens and 400 output tokens (with slight differences between various methods). For data synthesis using GPT-4, the cost of synthesizing one data point is calculated as:

$$10 \times 0.001 + 30 \times 0.0004 = 0.022\,\$$$

---

[‡]https://openai.com/api/pricing/

[§]https://power.netmind.ai/rentIntro

In terms of 0.01 cents, the synthesis cost is 220.

For data synthesis using GPT-3.5, the cost of synthesizing one data point is calculated as:

$$1.5 \times 0.001 + 2 \times 0.0004 = 0.0023 \,\$$$

In terms of 0.01 cents, the synthesis cost is 23.

For GRIP, we leveraged the vLLM [18] and used 8 NVIDIA A100 GPUs for 36 hours to construct 2.1 million data points. The cost of synthesizing one data point is calculated as:

$$\frac{0.42 \times 8 \times 36}{2123345} \approx 0.000057 \,\$$$

In terms of 0.01 cents, the synthesis cost is 0.57.

If we were to synthesize 2 million math problems and solutions, it would cost \$44000 using GPT-4, \$4600 using GPT-3.5, but only \$114 using GRIP. This gap becomes even more pronounced as the data volume increases.

## C    Benchmarks Overview

This section briefly introduces the datasets used in this paper, including the mathematical reasoning dataset, the scientific reasoning dataset, and the general ability dataset.

**MATH** [12]: MATH is a new dataset of 12,500 challenging competition mathematics problems. Each problem in MATH has a full step-by-step solution which can be used to teach models to generate answer derivations and explanations.

**GSM8K** [7]: This test dataset contains 1.32K diverse grade school math problems, intended to test basic arithmetic and reasoning ability in an educational context.

**GAOKAO-Eval** [32]: GAOKAO-Eval is a benchmark from China's Gaokao exam, covering various subjects and question types. Questions include multiple-choice, problem-solving, reading comprehension, and essay writing, with subjective answers scored by high school teachers. This paper evaluates only the mathematics test.

**SVAMP** [24]: SVAMP is a challenge set for evaluating models on elementary-level Math Word Problems (MWP). The dataset contains a total of 1,000 problems. Each MWP consists of a short natural language narrative that describes a state of the world and poses a question about some unknown quantities.

**ARC-C** [6]: ARC includes questions derived from various grade-level science exams, testing models' ability to handle both straightforward and complex scientific queries. We use the challenge subset, which contains 1,172 test questions.

**MMLU-STEM** [13]: Spanning 57 subjects across multiple disciplines, MMLU evaluates the breadth and depth of a model's knowledge in a manner akin to academic and professional testing environments. We select the STEM subset of MMLU with 3.13K problems.

**GPQA-Diamond** [26]: This dataset provides "Google-proof" questions in biology, physics, and chemistry, designed to test deep domain expertise and reasoning under challenging conditions. We use the diamond subset containing 198 hard problems.

**BIG-Bench Hard (BBH)** [28]: Consisting of 23 tasks previously found challenging for language models from BIG-Bench (Srivastava et al., 2023), BBH contains a total of 6511 challenging problems examining the capability of LLMs to solve them.

**TheoremQA** [5]: Focused on applying mathematical theorems to solve advanced problems in fields such as mathematics, physics, and engineering, TheoremQA includes 800 questions that test the theoretical reasoning capabilities.

**MBPP** [2]: MBPP consists of around 1,000 crowd-sourced Python programming problems, designed to be solvable by entry-level programmers, covering programming fundamentals, standard library functionality, and so on. Each problem consists of a task description, code solution, and 3 automated test cases.

**C-Eval** [16]: C-Eval is a comprehensive Chinese evaluation suite designed to assess the advanced knowledge and reasoning abilities of large language models. It includes multiple-choice questions across four difficulty levels (middle school, high school, college, and professional) and spans 52 diverse disciplines.

**MMLU** [13]: MMLU (Massive Multitask Language Understanding) is a benchmark that measures text models' multitask accuracy across 57 tasks, including elementary mathematics, US history, computer science, and law. It requires extensive world knowledge and problem-solving abilities, but even the best models still need significant improvements to reach expert-level accuracy.

# D Dual Filtering and KC Examples

## D.1 Dual Filtering

Ensuring the quality of KCs is crucial, as using meaningless KCs can result in low-quality synthesized problems while using overly similar KCs can lead to duplicated problems. These issues increase the computational and time costs for both problem synthesis and problem quality validation. We employ a dual filtering strategy using both embedding models and LLMs to remove low-quality and duplicated KCs. The three main steps are as follows:

**Eliminating Low-Quality KCs:** LLMs are used to filter out KCs that are vague, contain mathematical errors, or are overly detailed. This is because vague KCs can be too broad in meaning, failing to standardize the model's output effectively. Erroneous KCs may lead the model to synthesize incorrect questions, while overly detailed KCs can overly constrain the model's output.

**Categorization:** We first use an embedding model to calculate pairwise similarity scores between KCs. KCs with similarity scores between 0.90 and 1.0 are deemed to have the same meaning, while those with scores between 0.70 and 0.90 undergo an additional check by the LLM to confirm if they are truly synonymous. KCs with scores below 0.70 are treated as distinct. Based on this process, KCs are grouped into classes with similar KCs placed in the same class. These thresholds were determined through an analysis of the KC set.

**Summarization:** For each KC class, the LLM identifies the most representative KC to act as the class representative. If no existing KC in the class is suitable, the LLM synthesizes a new KC to represent the class. Finally, we obtained 10K qualified key concepts.

When only the embedding model was used for de-duplication, the quality check revealed that only 26% of the synthesized problems met the quality standard. After introducing dual filtering with the LLM, this proportion increased to 45%. This demonstrates that the dual filtering process significantly improves dataset quality while reducing problem synthesis costs.

## D.2 Examples of Bad Key Concepts

The LLM helps the embedding model classify KCs that appear similar but actually have different meanings. For example:

- *"Geometric sequence"* vs. *"Arithmetic sequence"* (similarity score: 0.805)
- *"Sine function in trigonometry"* vs. *"Cosine function in trigonometry"* (similarity score: 0.865)

The LLM effectively removes vague, mathematically incorrect, or overly detailed KCs. For example:

- Vague KCs:
  - *"Problem-solving strategies"*
  - *"Mathematical techniques"*
- Mathematically Incorrect KCs:
  - *"The sum of the outer angles of a polygon depends on the number of sides"*
  - *"The matrix result of multiplying a matrix by its inverse is the matrix itself"*
  - *"A series converges if its terms approach zero."*
  - Some incorrect or incomplete KCs

- Overly Detailed KCs:
  - *"Solving the quadratic equation $x^2 + 5x + 6 = 0$ by factoring…"*
  - Some specific problems

## D.3 Examples of Key Concepts

To demonstrate the diversity and comprehensiveness of our knowledge base, we randomly sampled 20 KCs:

*"Angle of Rotation", "The unit circle and its properties", "Solving Equations with Multiple Variables", "Right triangles in a sphere", "Inversions in permutations", "Pi ($\pi$) as a constant in geometry and trigonometry", "Perfect Cubes", "Area of Triangles and Squares", "Diophantine Approximation", "Perimeter of a triangle", "Abundant Number", "Graphing a hyperbola", "Determining the base and height of a Parallelogram", "Difference of cosines formula", "Quartic Polynomial", "Polynomial Inequalities", "Congruence of Integers", "Solving equations involving digits", "Sign Analysis", "Calculation of expected value for a fair eight-sided die".*

# E  Additional Experiments and Analyses

## E.1  Performance on Additional Challenging Benchmarks

To further reasonably demonstrate the performance gains of GRIP on more difficult benchmarks, we have already added some relatively challenging test benchmarks (e.g., GPQA-Diamond, TheoremQA) to Table 8 in the paper, where GRIP shows significant performance improvements compared to the base models. Furthermore, to further evaluate the models' ability to solve particularly complex mathematical problems, we have also introduced the AIME 2024 dataset and tested their pass@64 performance. These results, particularly on the notoriously difficult AIME benchmark, show a promising signal that GRIP can enhance complex reasoning capabilities, even if the absolute performance remains a frontier challenge. This improvement from zero demonstrates GRIP's potential to unlock new abilities in base models.

Table 8: Performance on additional challenging benchmarks. Results for GPQA-Diamond and TheoremQA are also presented in the main text, but are included here for a comprehensive comparison with AIME 2024.

| Model | Base | Size | GPQA-Diamond (Acc) | TheoremQA (Acc) | AIME 2024 (pass@64) |
|-------|------|------|--------------------|-----------------|---------------------|
| Mistral | - | 7B | 24.7 | 19.2 | 0/30 |
| LLaMA3 | - | 8B | 27.2 | 20.1 | 0/30 |
| Qwen2 | - | 7B | 32.3 | 33.5 | 3/30 |
| GRIP | Mistral | 7B | 31.7 | 26.5 | 4/30 |
| GRIP | LLaMA3 | 8B | 30.8 | 24.2 | 3/30 |
| GRIP | Qwen2 | 7B | **33.4** | **34.8** | **6/30** |

## E.2  Decontamination Analysis

To ensure the integrity of our results and address potential data contamination from benchmark test sets, we conducted a thorough decontamination analysis. Our synthesis method, GRIP, is fundamentally designed to generate novel problems by modeling key concepts and performing multi-hop combinations, rather than rephrasing or imitating existing examples. This design theoretically minimizes the risk of direct duplication. As demonstrated in our main analysis Table 1, our synthetic data exhibits low similarity to its seed data and achieves a high Novelty Rate of 71.8%.

To quantitatively verify the novelty of our dataset against standard benchmarks, we performed a formal n-gram overlap analysis between our `GRIP-MATH` training set and the official MATH test set. After normalizing the text of both datasets by lowercasing and removing punctuation, we calculated the percentage of overlapping n-grams for various lengths of n.

The results, presented in Table 9, show that the n-gram overlap rate is extremely low, particularly for longer sequences, indicating a negligible level of verbatim contamination. For instance, the 10-gram overlap is only 0.63%, and it drops to less than 0.01% for 15-grams.

Table 9: N-gram overlap analysis between the GRIP-MATH training set and the MATH test set.

| Dataset | N=8 | N=10 | N=13 | N=15 |
|---------|-----|------|------|------|
| MATH | 1.94% | 0.63% | 0.06% | <0.01% |

Furthermore, a qualitative analysis of the most frequent overlapping n-grams reveals that they consist of common mathematical phrases, definitions, or generic question stems, rather than specific problem content that would suggest data leakage. The top five most frequent overlapping sequences are:

*"What is the smallest possible value of the"*
*"How many zeros are at the end of"*
*"digit is the same as the units digit"*
*"digit of the sum of the squares of"*
*"is the sum of the lengths of these"*

In conclusion, this two-part analysis, combining our method's theoretical design with a direct empirical decontamination check, confirms that our synthesis process effectively avoids significant contamination from the benchmark test sets, thereby ensuring the validity of our evaluation results.

### E.3 Validation of Knowledge Concept Adherence

A crucial aspect of our data synthesis pipeline is its ability to generate problems that faithfully adhere to the guiding knowledge concepts (KCs). To quantitatively evaluate this, we conducted a reverse-validation experiment. Specifically, we employed the same concept extraction model used in our initial pipeline (Qwen2.5-32B-Instruct) to perform a "reverse" concept extraction on a sample of the synthesized problems. Subsequently, we calculated the match ratio between the newly extracted concepts and the original concepts that were used to generate these problems.

We define two levels of adherence to measure the fidelity of the synthesis process. A **Full Match** occurs if all of the original input concepts were successfully re-extracted from the generated problem. A **Partial Match** is registered if at least one of the original input concepts was found.

Table 10: Adherence ratio of synthesized problems to the guiding knowledge concepts.

| Adherence Level | Adherence Ratio |
|-----------------|-----------------|
| Full Match | 88.65% |
| Partial Match | 98.49% |

The results, shown in Table 10, demonstrate a very high degree of fidelity. The Full Match ratio of 88.65% confirms that our method is highly effective at integrating all specified concepts into a coherent problem. Furthermore, the 98.49% Partial Match ratio indicates that even in cases where a perfect combination is not achieved, the generated problems remain strongly relevant to the intended knowledge domains. This validation confirms that our data synthesis process is not only scalable but also precise in following the conceptual guidance provided by the knowledge graph.

### E.4 Performance Analysis Based on Dataset Scale

To provide a more granular analysis of our method's performance, we present a supplementary comparison that evaluates various data synthesis methods on a single model architecture (Mistral-7B), with datasets grouped by their approximate scale. The effectiveness of a data synthesis method is typically evaluated on two key dimensions: the *quality* of the generated data and the *scalability* of the synthesis process. This analysis is designed to offer further insight into GRIP-MATH's performance along both of these dimensions. The results are presented in Table 11, which is divided into two categories: datasets with approximately 100K samples and those with over 1 million samples.

**Performance at a Smaller Scale (~100k).** To facilitate a direct comparison of data quality against datasets of a similar size, we randomly sampled 80K question-answer pairs from our full dataset to create GRIP-MATH-mini. As shown in Table 11, this allows for a controlled evaluation where data quantity is normalized. The results demonstrate that the superior quality of data generated by our synthesis method leads to better model performance. Notably, a model trained on GRIP-MATH-mini (43.0%) significantly surpasses one trained on MathCoder (38.8%), which has the same number of samples.

Table 11: Performance comparison of various data synthesis methods on the Mistral-7B model, grouped by dataset scale. The 'AVG' column refers to the average score on the mathematical reasoning benchmarks detailed in the main paper.

| Scale | Dataset | Total Seed Data | Total Synthesized Data | AVG |
|---|---|---|---|---|
| *~100K* | MetaMath | 15K | 395K | 40.3 |
| | WizardMath | 15K | 96K | 38.0 |
| | MAmmoTH | 220K | 262K | 39.0 |
| | MathCoder | 15K | 80K | 38.8 |
| | **GRIP-MATH-mini** | **7.5K** | **80K** | **43.0** |
| *>1M* | MathScale | 20K | 2M | 51.2 |
| | MAmmoTH2 | >10M | 10M | 52.2 |
| | **GRIP-MATH** | **7.5K** | **2.1M** | **57.3** |

**Performance at a Larger Scale (>1M).** The second part of the analysis highlights the key metric of scalability. When comparing our full 2.1M-sample dataset against other large-scale methods, GRIP-MATH demonstrates superior efficiency and quality. For example, compared to MAmmoTH2, which filters 10M samples from the massive Common Crawl corpus, our method achieves a 5.1-point higher score (57.3% vs 52.2%) using only 7.5K seed examples. Similarly, our method outperforms MathScale (which uses GPT-3.5) by a margin of 6.1 points. This analysis proves that our synthesis method not only scales data volume effectively but also ensures superior data quality at scale.

# F   Limitations and Future Work

The primary limitation of this work is that the empirical validation of the GRIP framework is confined to the domain of mathematical reasoning. While our results demonstrate significant success within this area, the effectiveness of GRIP in other domains has not yet been verified.

Our decision to initially focus on mathematics was deliberate. Mathematical reasoning represents a significant challenge for Large Language Models (LLMs), and success in this domain provides strong evidence of a data synthesis method's efficacy. Furthermore, the well-defined, hierarchical knowledge structure of mathematics offered an ideal environment to validate our core hypothesis: that novel and complex problems can be synthesized by combining foundational knowledge concepts.

Despite this specific focus, our work provides initial evidence of GRIP's broader applicability. As shown in Table 3, models trained exclusively on GRIP-MATH demonstrate improved performance on scientific reasoning benchmarks (e.g., BBH, GPQA-Diamond, MMLU-STEM), indicating a degree of cross-domain generalization. We posit that the core GRIP pipeline—"Concept Extraction → Knowledge Graph Construction → Concept Combination → Data Synthesis → Filtering"—is fundamentally domain-agnostic. The key to extending GRIP to new domains lies in adapting the definition of a "concept" to the target domain. We envision two primary directions for this extension:

- **For structured domains** like STEM and programming, the mapping is straightforward, as "concepts" can be directly defined as clear rules, principles, or library functions.

- **For less-structured domains** like commonsense reasoning, we hypothesize this paradigm remains effective. Here, "concepts" can be defined as higher-level scenarios or activities. For instance, GRIP could combine two distinct but related concepts, such as *"planning an international trip"* and *"dealing with a lost passport,"* to create a novel, complex reasoning problem that leverages their implicit relationships.

Therefore, a top priority for our future research is to rigorously extend and validate the GRIP framework in these diverse domains, including commonsense and procedural reasoning. Verifying its effectiveness in these areas will be crucial for establishing GRIP as a truly general-purpose data synthesis methodology.

