# OpenReview forum: "GRIP: A Graph-Based Reasoning Instruction Producer"
_NeurIPS.cc/2025/Conference — NeurIPS 2025 poster_

### Official Review · Reviewer_Juc8 · 2025-06-21

**Clarity:** 3
**Significance:** 4
**Originality:** 3
**Rating:** 4
**Confidence:** 3

**Summary:**

GRIP successfully addresses the limitations of existing data synthesis methods in terms of scalability, diversity, and cost-effectiveness through its innovative graph-based approach. This method not only achieves significant results on mathematical reasoning tasks but also demonstrates strong cross-domain generalization capabilities, providing a new paradigm for data synthesis in large language models.

**Questions:**

1. Line 189 mentions using a rating model to obtain problem difficulty. What specific model is used and how is it implemented?
2. When generating data, multiple open-source models are used to jointly score problems and answers, directly providing absolute score values. The article does not explain the method in detail - is it done by directly prompting the open-source models to give scores? If scoring directly, do open-source models truly possess the capability similar to reward models to directly assign definitive scores to problems and answers?
3. The article validates diversity by comparing the similarity distribution between generated data and seed data, but does not provide specific calculation methods. For example, GRIP generates through key concepts without directly corresponding seed data, while some other methods have directly corresponding data. Is the similarity calculated for all data, or only for corresponding parts, or taking the maximum value? Supplementing calculation details would make the results more convincing.

**Ethical Concerns:**

["Major Concern: Data quality and representativeness"]

**Final Justification:**

I am satisfied with the author’s response and keep my score unchanged.

**Limitations:**

The article compares multiple data synthesis methods, but different methods use different source data, with significant variations in the amount of generated data, all of which are lower than the proposed method. On one hand, the ability to synthesize such a large amount of high-quality data is indeed an advantage of the proposed method, which deserves recognition; on the other hand, if more fair comparison results could be provided under the premise of controlling for the same source data and generated data volume, it would undoubtedly further demonstrate the effectiveness of the proposed method.

**Quality:**

4

**Strengths And Weaknesses:**

**Strength**
1. This is an interesting work with a very clever idea (I have had similar thoughts before, and I'm glad you implemented and validated its effectiveness).
2. A relatively large-scale mathematical dataset has been constructed, which is very meaningful for model training and evaluation. I believe this method also has certain scalability.
3. The overall writing of the paper is relatively clear, although I still have some questions after reading it.

**Weakness**
1. Do the synthesized problems truly adhere to these knowledge concepts? I think it's necessary to perform knowledge concept extraction operations again to check whether the generated problems follow the provided knowledge concepts, and what is the adherence ratio?
2. Specific questions are detailed in the "Questions" section.

**Typos**
1. There is a reference error on line 130.

---

> ### Author Rebuttal · Authors · 2025-07-31
>
> ## Validation of Knowledge Concept Adherence
>
>
> Thank you for your highly insightful suggestion to "reverse-validate" whether the generated problems adhere to the given knowledge concepts.
>
> We have adopted your suggestion and conducted this experiment. Specifically, we used the knowledge concept extraction model mentioned in our paper (Qwen2.5 32B) to perform "reverse" concept extraction on our synthesized problems. Subsequently, we calculated the match rate (i.e., the "adherence ratio" you mentioned) between the newly extracted concepts and the original concepts that were used to generate these problems. We define two levels of adherence:
>
> - **Full Match:** All of the original input concepts were successfully extracted from the generated problem.
> - **Partial Match:** At least one of the original input concepts was successfully extracted from the generated problem.
>
> |               | Adherence Ratio |
> | ------------- | --------------- |
> | Full Match    | 88.65%          |
> | Partial Match | 98.49%          |
>
> These results demonstrate that our data synthesis process faithfully adheres to the guidance of the given knowledge concepts, ensuring the relevance and accuracy of the generated content. The high "Partial Match" rate also indicates that the generated problems remain highly relevant to the intended knowledge domains even when a perfect combination isn't achieved. We will add this crucial validation experiment to the revised version of our paper. Thank you again for your valuable feedback.
>
>
>
> ## Specification of the Problem Difficulty Rating Model
>
> Thank you for the question. We will provide a more detailed description in the revised manuscript. The model mentioned is an **internal model** that we developed, based on the BERT architecture. It was fine-tuned on a proprietary internal dataset. On this dataset, problems were assigned difficulty labels (easy, medium, or hard) through a combination of our own **in-house human annotation** and generation assisted by ChatGPT. Fine-tuning on our unique internal data enables the model to assign difficulty labels to new problems.
>
> ## Methodology and Reliability of Multi-Model Quality Scoring
>
> Thank you for your question, as it touches upon a core mechanism of our methodology. We designed a **Structured Evaluative Prompt**. By clearly defining the dimensions for evaluation and providing a few human-annotated samples as few-shot examples, we guide the models to perform a multi-dimensional, quantifiable assessment, rather than simply having them assign an absolute score directly.
>
> As described in Section 3.4 of the paper, for the evaluation of **problems**, we focus on two core criteria: **Logical Completeness** and **Presentational Completeness**. These are designed to assess whether a problem contains mathematical errors or logical contradictions, whether it is closely related to the given knowledge concepts, and whether its phrasing is complete and clear. We then guide the model to produce a final score by enforcing a structured output.
>
> For the evaluation of **solutions**, we employ a similar but stricter strategy. We also guide the models to assess the solution's quality via prompts. However, for higher accuracy, we first have the model generate a detailed reasoning process and then provide a binary "correct" or "incorrect" judgment based on that reasoning for the purpose of **rejection sampling**. Furthermore, GRIP implements a **"unanimous consent" mechanism**, where a solution is only accepted if all evaluation models unanimously judge it as correct. This greatly enhances data accuracy.
>
> Regarding your second, and more central, concern—**"Do open-source models truly possess this capability?"**—we provide an affirmative answer through a key ablation study.
>
> In the ablation study in Section 4.5, we conducted an experiment specifically to validate the reliability of our multi-model evaluation framework. We randomly sampled 500 synthetic math problems and had human experts manually label them for quality ("qualified" or "unqualified"). Using this human-labeled set as the ground truth, we tested the accuracy of our three-model evaluation framework (and GPT-4.1 for comparison). As shown in **Table 5**, the collaborative evaluation accuracy of our three open-source models reached **95.7%**, a performance that even surpassed the accuracy of using GPT-4.1 alone for evaluation **(94.3%)**.
>
> This result strongly demonstrates that through our designed framework—which is multi-model, multi-dimensional, and uses structured outputs—the open-source models, when used in concert, can indeed make highly reliable judgments on the quality of problems and answers. Their effectiveness is comparable to, and can even exceed, that of top-tier closed-source models. Additionally, regarding your point on reward models, some studies have already shown that generative reward models [1,2] can also achieve strong performance.
>
> [1] Mahan D, Van Phung D, Rafailov R, et al. Generative reward models[J]. arXiv preprint arXiv:2410.12832, 2024.
>
> [2] Li H, Dong Q, Chen J, et al. Llms-as-judges: a comprehensive survey on llm-based evaluation methods[J]. arXiv preprint arXiv:2412.05579, 2024.
>
> ## Clarification on the Diversity Metric Calculation Method
>
> Thank you for your question regarding the calculation details for our diversity metrics. To ensure an absolutely fair comparison, we applied **an identical and uniform evaluation protocol** to all methods. For each method, we used its own respective seed data and synthesized data for the analysis.
>
> **Similarity Score Calculation (for Figure 4):**
>
> In Table 1, we list the source and scale of the seed data for all data synthesis methods. To calculate the similarity, we used the **BGE m3-embedding model** for each respective seed dataset.
>
> For each individual problem in a synthetic dataset, we calculate its embedding similarity against **all** problems in its corresponding seed dataset. We then take the **maximum value** from these comparisons as the final similarity score for that synthesized problem. The distributions of these maximum scores are shown in Figure 4.
>
> The calculation for the "Novelty Rate" metric is defined as: "the percentage of problems whose combination of knowledge points did not appear in the seed dataset."
>
> We will update the next version of our paper to include these specific calculation details to make our results more convincing.
>
> ## Fair comparison
>
> We have reorganized the experiments in Table 2 to improve readability and fairness.
> |        | Datasets             | Seed Dataset Count | Dataset Count | Avg      |
> | :----- | :----------------- | :----------------- | :------------ | :------- |
> | ～100K | MetaMath           | 15K                | 395K          | 40.3     |
> |        | WizardMath         | 15K                | 96K           | 38.0     |
> |        | MAmmoTH            | 220K               | 262K          | 39.0     |
> |        | MathCoder          | 15k                | 80K           | 38.8     |
> |        | **GRIP-MATH-mini** | **7.5k**           | **80k**       | **43.0** |
> | >1M    | MathScale          | 20K                | 2M            | 51.2     |
> |        | MAmmoTH2           | >10M               | 10M           | 52.2     |
> |        | **GRIP-MATH**      | **7.5k**           | **2.1M**      | **57.3** |
>
> - **Performance at a Smaller Scale (~100k)**
>
> To directly address your point about comparing datasets of the "same number," we randomly sampled 80k Q-A pairs from our synthesized data to create **GRIP-math-mini**. This allows for an apples-to-apples comparison with datasets of a similar size, like MathCoder (80k). The results show that even when controlling for data quantity, our synthesis method's superior quality leads to better performance, outperforming existing methods in this category. For instance, **GRIP-math-mini (43.0%)** significantly surpasses **MathCoder (38.8%)**.
>
> - **Performance at a Larger Scale (>1M)**
>
> This comparison highlights the second key metric: scalability. When comparing our full 2.1M dataset against other large-scale methods, GRIP-math demonstrates superior efficiency and quality. For example, compared to MAmmoTH2, which filters 10M samples from the massive Common Crawl, our method achieves a 5.1% higher score (57.3% vs 52.2%) using only 7.5k seed examples. Similarly, our method outperforms MathScale (which uses GPT-3.5) by 6.1%. This proves that our synthesis method not only scales data volume effectively but also ensures superior data quality.
>
> We will update the manuscript to include this detailed table and analysis. This will make the evaluation more transparent, comprehensive, and directly address your valid concerns about fairness. Thank you again for this valuable suggestion.
> ## Paper Formatting Concerns:
> Thank you for your careful review and for pointing out the formatting error on Line 130. This was indeed a LaTeX label rendering issue where "Appendix ??" was displayed incorrectly. We have now corrected this cross-reference, and it correctly points to Appendix D in the revised manuscript. We appreciate your valuable feedback.

---

> > ### Comment · Reviewer_Juc8 · 2025-08-05
> >
> > Thanks for the detailed response. I am OK for the work if they can incorporate all these comments into their final paper.

---

> > > ### Author Response · Authors · 2025-08-05
> > > **To Reviewer Juc8**
> > >
> > > Thank you very much for your positive feedback. We are very pleased to hear that we have successfully addressed your concerns. We solemnly commit to fully incorporating all of your valuable comments into the final version of the manuscript. Thank you again for your time and guidance.

---

### Official Review · Reviewer_Bfeb · 2025-06-23

**Clarity:** 2
**Significance:** 3
**Originality:** 3
**Rating:** 4
**Confidence:** 3

**Summary:**

This paper introduces **GRIP** (Graph-based Reasoning Instruction Producer), a novel framework for synthesizing large-scale, high-quality, and diverse mathematical reasoning data. In contrast to traditional data synthesis approaches that rephrase or augment seed examples, GRIP departs from direct manipulation of seed data. Instead, it constructs a knowledge graph from extracted key concepts and leverages both explicit and implicit relationships within the graph to synthesize new questions. A multi-model supervision mechanism is employed to filter the generated data, ensuring quality without relying on expensive closed-source models. Applied to the MATH dataset, GRIP synthesizes a 2.1M dataset (GRIP-MATH) from 7.5K seeds, achieving a 280× expansion while significantly reducing generation cost. Empirical evaluations show that models trained on GRIP-MATH outperform prior data synthesis baselines and even rival proprietary math-specialist models in both mathematical and scientific reasoning tasks.

**Questions:**

1. Figures 1 and 2 are difficult to read due to small font and dense layout. Could the authors improve the resolution, enlarge the key annotations, or separate parts of the diagrams for better interpretability?

2. Lines 45–57 discuss the scalability benefits of using non-co-occurring concept pairs. Is there any empirical analysis or ablation study that directly supports this observation (e.g., concept pair count vs. effective data generation rate)?

3. The current evaluation is limited to mathematical reasoning tasks. Could the authors comment on how GRIP might generalize to other domains (e.g., science, programming, commonsense QA), possibly by leveraging semantic graphs or domain-specific ontologies?

4. The construction process of the KCRG appears heuristic. Is there a formally defined algorithm or pseudo-code for how the graph is constructed, especially for identifying and weighting implicit relationships?

5. The evaluation primarily focuses on benchmark accuracy, but lacks qualitative insights.
   Could the authors include a small-scale human evaluation or case study to highlight strengths and potential failure modes in the generated data?

**Ethical Concerns:**

["NO or VERY MINOR ethics concerns only"]

**Final Justification:**

After carefully considering the author’s rebuttal, the AC’s input, and the points raised in the other reviewers’ discussions, I find that my original assessment remains valid. While the rebuttal addressed certain concerns, it did not sufficiently resolve the key issues I raised. Therefore, I will keep my original score.

**Limitations:**

yes

**Paper Formatting Concerns:**

No.

**Quality:**

3

**Strengths And Weaknesses:**

## Strengths

1. The paper conducts comprehensive experiments across multiple benchmarks and model architectures, with detailed ablations that validate the effectiveness and scalability of the proposed method.

2. GRIP introduces a novel graph-based synthesis strategy that systematically combines both explicit and implicit concept relationships, enabling diverse and large-scale generation beyond prior methods.

## Weaknesses

1. The method is only evaluated on mathematical data, and its applicability to domains lacking structured concept representations (e.g., commonsense reasoning) remains unverified.

2. Data quality tends to degrade as the graph hop distance increases, which may reduce the reliability of more diverse combinations even with filtering.

3. The evaluation focuses primarily on quantitative metrics, with limited qualitative analysis of synthesized examples, making it difficult to assess failure modes or interpretability aspects.

---

> ### Author Rebuttal · Authors · 2025-07-31
>
> ## Generalization to Other Domains
>
> Thank you for this insightful comment. We agree that the generalization of GRIP is a crucial point. We would like to address this from three perspectives:
>
> **Rationale for Starting with Mathematics**: Following the existing works, our experiments with GRIP are focused on the domain of mathematical reasoning. We chose mathematics as the initial application area for two primary reasons. First, mathematical reasoning is one of the most challenging tasks for LLMs; success in this domain provides strong evidence of a data synthesis method's efficacy. Second, the well-defined, hierarchical knowledge structure of mathematics offered an ideal environment to validate our core hypothesis: generating novel and complex problems by combining foundational knowledge concepts.
>
> **Generalization Potential of the GRIP Framework:** While mathematics was our primary focus, our work already provides initial evidence of GRIP's broader applicability. As shown in **Table 3**, models trained on GRIP-MATH demonstrate improved performance on scientific reasoning benchmarks (e.g., **BBH, GPQA, MMLU-STEM**), indicating cross-domain generalization.
>
> We posit that the core GRIP pipeline—**“Concept Extraction → Knowledge Graph Construction → Concept Combination → Data Synthesis → Filtering”**—is fundamentally domain-agnostic. The key is adapting the **definition of a "concept"** to the target domain.
>
> - **For structured domains** like STEM and programming:  the mapping is straightforward, as they rely on clear rules and principles.
>
> - **For less-structured domains** like commonsense reasoning, we hypothesize this paradigm remains effective. Here, "concepts" can be defined as higher-level scenarios or activities. For instance, GRIP could combine two distinct but related concepts, such as **"planning an international trip"** and **"dealing with a lost passport."** Their combination creates a novel, complex reasoning problem by leveraging implicit relationships, which is GRIP's core strength.
>
> **Acknowledged Limitations and Future Work:** We are transparent about the scope of our current experiments. As stated in our conclusion (**Section 5**), *"Intuitively, GRIP should be applicable to various domains... however, our current experiments are limited to the mathematics domain, and its effectiveness in other areas is yet to be verified."* Extending and rigorously validating the GRIP framework in diverse domains, including commonsense and procedural reasoning, is a top priority for our future research.
>
> ## Data Quality and Hop Distance
>
> Thank you for this insightful comment. We agree and have also observed that data quality tends to decrease with greater hop distances. This is precisely why we **integrated specific strategies** to mitigate this effect while still harnessing the **diversity from implicit relationships**.
>
> **Direct Observation**: Our ablation study in **Table 4** confirms this trend. The average quality score drops from **0.94 (one-hop)** to **0.72 (two-hop)** and **0.62 (three-hop)** as distance increases.
>
> **Mitigation Strategies**: To ensure the reliability of more distant combinations, we implemented two key strategies:
>
> - **Hub Concept Prioritization**: For three-hop relationships, we exclusively generate combinations involving a **hub concept**—a central, highly-connected node in our knowledge graph. This maintains semantic relevance even across longer distances.
> - **Weight-Based Filtering**: We apply stricter filtering to multi-hop combinations, removing low-weight (i.e., less frequent or relevant) pairs to ensure the selected concepts are information-rich.
>
> **Effectiveness of Strategies**: The effectiveness of the hub concept strategy is validated in **Table 4**, which shows that three-hop combinations generated *with* our hub concept strategy achieve a score of **0.62**, whereas those generated *without* it perform significantly worse at just **0.39**.
>
> **Performance evidence**: Most importantly, the final model performance demonstrates the value of this curated diversity. As shown in **Table 7**, training on a dataset that includes these carefully filtered two-hop and three-hop implicit combinations leads to a notable improvement on the GSM8K benchmark (score improves from 67.6 to **71.2**) compared to using explicit data alone.
>
>
>
> ## Qualitative Analysis of Synthesized Data
>
> Thank you for your valuable suggestion. In the revised manuscript, we will add a new section to the appendix. This section will showcase concrete question-and-answer samples generated from different types of concept combinations (one-hop, two-hop, and three-hop), along with a brief analysis of them.
>
> ## Empirical Support for Scalability
>
> Thank you for your advice. We conducted an empirical analysis to precisely measure the scalability benefits and data contribution of using non-co-occurring (multi-hop) concept pairs. The results directly support our observation.
>
> The analysis confirms that these multi-hop, non-co-occurring pairs are the primary driver of both scalability and final data volume. The number of available concept pairs expands dramatically when moving from direct co-occurrence to implicit, multi-hop relationships. While direct 1-hop pairs are highly effective, their limited number means they contribute only **16.64%** of the final dataset. In contrast, the non-co-occurring **2-hop and 3-hop pairs collectively generate over 71%** (49.85% + 21.98%)of the total effective data.
>
> |            | Combination Count | Effective Data Generated | Effectiveness Rate(%) | % of Total Data |
> | ---------- | ----------------- | ------------------------ | --------------------- | --------------- |
> | one-hop   | 450663            | 353123                   | 78.4                 | 16.6           |
> | two-hop   | 1771511           | 1058133                  | 59.7                 | 49.9           |
> | three-hop | 315744            | 466719                   | 47.6                 | 22.0           |
> | Community  | 285744            | 244485                   | 77.4                 | 11.5           |
>
> ## Formalizing the Graph Construction Algorithm
>
> Thank you for this important question. We appreciate the opportunity to clarify that the KCRG construction is a **deterministic, algorithm-driven process**, not a heuristic one. The core logic is detailed in Section 3.2. We are happy to outline the precise algorithmic steps below and will add this clarification to the final version of the paper.
>
> The construction process follows these formal steps:
>
> **Step 1: Node Set (K) Construction**
>
> 1. After extracting Key Concepts (KCs) from seed data, we apply a rigorous dual-filtering process to create a unique set of high-quality nodes, K.
>
> **Step 2: Explicit Edge (E_ex) Construction & Weighting**
>
> 1. **Data Structuring:** After extracting all KCs, we store their occurrences in a key-value format (e.g., JSONL). Each line maps a KC to a list of problem IDs where it appears.
>
>    ```
>    {
>      "key_concept1": [1, 2, 8, 15],
>      "key_concept2": [2, 8, 22, 31, 57]
>      ...
>    }
>    ```
>
> 2. **Identification:** An explicit (one-hop) edge `(k_i, k_j)` is created between two KC nodes if their problem ID lists have a non-empty intersection.
>
> 3. **Weighting:** The weight of an explicit edge `W(k_i, k_j)` is strictly defined as the **count of co-occurring problems** (the size of the intersection of their problem ID lists). For instance, in the example above, "key_concept1" and "key_concept2" co-occur in `problem_2,problem_8`. If this is their only co-occurrence, the weight of the edge between them is 2.
>
> **Step 3: Implicit Relationship Identification & Weighting** Our method explores implicit relationships by identifying multi-hop paths. Crucially, we use the weights of the constituent explicit edges to filter for high-quality connections.
>
> 1. **2-Hop Path Identification:** A 2-hop path is identified between nodes `(k_i, k_m)` if their shortest path distance is exactly two, i.e., `D(k_i, k_m) = 2`. These paths represent concept pairs sharing a common neighbor.
> 2. **3-Hop Path Identification (Hub-Constrained):** A 3-hop path is identified between `(k_i, k_m)` if `D(k_i, k_m) = 3`, under the strict condition that the path must involve at least one **Hub Concept** .
>    - A Hub Concept is formally defined as a node whose degree falls within the **top 5%** of the entire graph's degree distribution.
> 3. **Weighting for Implicit Relationship **: The weight of an implicit path is the ***minimum* weight** of any explicit edge along it. We filter out low-weight implicit paths to ensure a baseline of semantic coherence.
>
> Here is the pseudo-code
>
> ```
> Algorithm 1: KCRG Construction
> ---------------------------------------------------------------------
> Input: Seed dataset D, Hub concept degree threshold k
> Output: Graph G = (K, E_ex, W_ex, E_im)
>
> // Step 1: Node Set Construction
> K_raw ← {}
> for problem p in D do
>   K_p ← ExtractKeyConcepts(p)
>   K_raw ← K_raw ∪ K_p
> K ← DualFilter(K_raw)
>
> // Step 2: Explicit Edge Construction & Weighting
> G ← InitializeGraph(Nodes=K)
> Co-occurrence_Map ← BuildProblemIDMap(D, K)
> for k_i, k_j in pairs(K) where i < j do
>   Intersection_IDs ← Co-occurrence_Map[k_i] ∩ Co-occurrence_Map[k_j]
>   if |Intersection_IDs| > 0 then
>     weight ← |Intersection_IDs|
>     G.AddEdge(k_i, k_j, weight=weight, type='explicit')
>
> // Step 3: Implicit Relationship Identification
> Hub_Concepts ← FindNodesWithDegree(G) > k // Top-k percentile
> for k_i in K do
>   // 2-Hop
>   Neighbors_2_hop ← FindPaths(G, start=k_i, depth=2)
>   G.AddImplicitRelations(k_i, Neighbors_2_hop, type='2-hop')
>   // 3-Hop (Hub-Constrained)
>   Neighbors_3_hop ← FindPaths(G, start=k_i, depth=3, through=Hub_Concepts)
>   G.AddImplicitRelations(k_i, Neighbors_3_hop, type='3-hop')
>
> return G
> ```
> This rule-based process is reproducible and provides a solid foundation for our data synthesis. We will also release the code.
>
> ## Readability
> We will make revisions in the final version.

---

> ### Comment · Reviewer_Bfeb · 2025-08-03
>
> Thank you for the comprehensive and thoughtful responses. I appreciate the detailed clarifications and additional analyses provided, particularly regarding the generalization potential of GRIP, the handling of multi-hop relationships, and the formalization of the graph construction process.
>
> The authors have convincingly addressed my concerns.

---

> > ### Author Response · Authors · 2025-08-04
> > **To Reviewer Bfeb**
> >
> > Thank you very much for your positive feedback. We are pleased to hear that our revisions and clarifications have successfully addressed your concerns. Your valuable comments have been instrumental in improving the quality of our manuscript. We thank you again for your time and effort during the review process.

---

### Official Review · Reviewer_3rLC · 2025-07-02

**Clarity:** 4
**Significance:** 3
**Originality:** 3
**Rating:** 5
**Confidence:** 4

**Summary:**

The paper introduces a method of synthetic data generation that leverages concepts graphs to produce novel and diverse data. GRIP builds a key concept graph from seed data and then synthesizes new problems by prompting an LLM with concept combinations selected from this graph.

**Questions:**

* The paper does not mention data decontamination, which is a standard procedure for synthetic data generation. Could the authors perform and report the results of a standard n-gram decontamination analysis between synthetic dataset and all benchmark test sets? If significant overlap is found, please include a discussion on how this might affect the paper's conclusions.

* The paper would be strengthened by a more explicit justification for the choice of models and benchmarks for experiments. Given that benchmarks like GSM8K are largely considered solved by frontier models, could the authors please elaborate on why this setting was chosen?

**Ethical Concerns:**

["NO or VERY MINOR ethics concerns only"]

**Final Justification:**

I am satisfied with the authors’ clarifications; they resolve most of my concerns. I will update my score.

**Limitations:**

No. The authors have not included a limitations as a separate section and briefly mention them in the section 5. A key limitation that should be discussed is the choice of non-reasoning models and simple benchmarks for evaluation. An honest discussion would acknowledge that while the generated data shows promise on these models, its utility for state-of-the-art long-reasoning models remains unverified.

**Paper Formatting Concerns:**

At the end of section 3.1 the paper references non-existing appendix section.

**Quality:**

3

**Strengths And Weaknesses:**

## Strengths
* The paper is generally well-structured, with a clearly stated motivation and reasonable experiments and ablation studies.
* The described method is novel and makes a significant contribution to the field of synthetic data generation. In particular, its ability to generate diverse problems while avoiding the direct use of seed questions and the use of implicit concept connections makes the approach valuable.
* The authors' commitment to release their code and the 2.1M problem dataset is another important contribution.

## Weaknesses

* **Focus on the non-reasoning models**

    The evaluation is conducted on benchmarks (e.g., GSM8K, SVAMP) that are considered largely saturated and with models that are far behind the current frontier reasoning models, such as QwQ-32B and DeepSeek-R1. While the experimental validation of the method is thorough, its capabilities are not demonstrated on the more complex benchmarks (e.g. AIME24/25, HMMT24/25, GPQA Diamond) that are more conventional now. The lack of justification for this experimental design choice weakens the paper's overall claims.

* **Lack of decontamination**

    Even though the presented method does not directly rephrase seed problems, a standard concern for synthetic data generation is potential contamination with benchmark test sets. The paper does not provide any details regarding decontamination.

* **Incomplete cost comparison**

    The cost comparison in Table 1 does not include any open-source-based data generation methods. This leads to an incomplete comparison, as all listed baselines rely on proprietary models (e.g., GPT-3.5, GPT-4), making it difficult to fully assess GRIP's cost-effectiveness against non-proprietary alternatives.

---

> ### Author Rebuttal · Authors · 2025-07-31
>
> ## Justification of Benchmark and Model Choices in Evaluation
>
> Thank you for your insight on our experimental design choices. We chose benchmarks like GSM8K and MATH, as well as foundational models, based on the following considerations.
>
> **Evaluation consisency**：As shown in Table 2, We initially focused on benchmarks such as GSM8K and MATH using widely adopted foundation models (e.g., Mistral, Qwen, Llama) to maintain consistency with prior data synthesis studies (e.g., MathScale, MetaMath). These benchmarks are still broadly accepted in evaluating mathematical reasoning and allow fair comparison across methods under equivalent training conditions. Importantly, our goal was to isolate and validate the effectiveness of the GRIP data synthesis strategy, independent of model capabilities.
>
> **Evaluation on Challenging Benchmarks**:  To further reasonably demonstrate the performance gains of GRIP on more difficult benchmarks, we have already added some relatively challenging test benchmarks, e.g. GPQA TheoremQA, to Table 2 in the paper, where GRIP shows significant performance improvements compared to the base models. Furthermore, to further evaluate the model's ability to solve particularly complex mathematical problems, we have also introduced the AIME 2024 dataset and tested their pass@64 performance. These results, particularly on the notoriously difficult AIME benchmark, show a promising signal that GRIP can enhance complex reasoning capabilities, even if the absolute performance remains a frontier challenge. This improvement from zero demonstrates GRIP's potential to unlock new abilities in base models.
>
> |              | GPQA-Diamond（Acc） | TheoremQA（Acc） | AIME2024（pass@64） |
> | ------------ | ------------------- | ---------------- | ------------------- |
> | Mistral   | 24.7                | 19.2             | 0/30                |
> | GRIP-Mistral | 31.7                | 26.5             | 4/30                |
> | LLaMA3       | 27.2                | 20.1             | 0/30                |
> | GRIP-LLaMA3   | 30.8                | 24.2             | 3/30                |
> | Qwen2        | 32.3                | 33.5             | 3/30                |
> | GRIP-Qwen2   | 33.4                | 34.8             | 6/30                |
>
> **Complementary to reasoning-based approaches**: GRIP is designed as a general, scalable data synthesis framework that is **model- and benchmark-agnostic**. Our core contribution lies in enabling large-scale generation of diverse and high-quality reasoning data by systematically modeling concept-level relationships, rather than relying on specific CoT traces or frontier model outputs. This approach **complements existing methods** that improve reasoning through CoT distillation or reinforcement learning. While our current experiments focus on commonly used seed data and foundation models to ensure evaluation consistency, GRIP is fully compatible with stronger seeds and models. As part of ongoing work, we plan to apply GRIP to more advanced datasets (e.g., DeepScaleR) and frontier models such as DeepSeek-R1 and Qwen3-235B to assess its effectiveness under high-capacity reasoning settings. Due to rebuttal time constraints, we could not complete such experiments, but we will clearly acknowledge this in the paper’s “**Limitations**” section and treat it as a key direction for future work.
>
>
>
> ## Lack of decontamination
>
> Thank you for your valuable comment regarding data decontamination. While our paper indirectly demonstrated the novelty of our GRIP dataset compared to the seed data, we acknowledge that a direct analysis of potential contamination from benchmark test sets is necessary. Our synthesis method, which is based on modeling **key concepts and performing multi-hop combinations,** is fundamentally designed to generate new problems rather than rephrasing or imitating existing ones. This theoretically minimizes the risk of direct duplication. For example, the "two-hop" and "three-hop" problem structures generated by our method are largely absent from the original seed dataset. Our analysis in **Section 3.5 (Figure 4, Table 1)** already supports this, showing that our synthetic data exhibits **low seed similarity** and a **high Novelty Rate** of 71.8%.
>
> To directly address your concern about benchmark contamination, we conducted a formal n-gram overlap analysis between our GRIP-math training set and the MATH test set. After normalizing the text of both datasets, we calculated the overlap rate for various n-gram lengths. The results are as follows:
>
> | N    | 8     | 10    | 13    | 15     |
> | ---- | ----- | ----- | ----- | ------ |
> | Math | 1.94% | 0.63% | 0.06% | <0.01% |
>
> The results clearly show that the n-gram overlap rate between the GRIP-math training set and test benchmark is extremely low, especially for longer sequences, indicating a negligible level of verbatim contamination. Furthermore, a qualitative analysis of the most frequent overlapping n-grams reveals that they consist of common mathematical phrases, definitions, or generic question stems, rather than specific problem content. Here are the top 5 overlaps:
>
> ```
> "What is the smallest possible value of the"
> "How many zeros are at the end of"
> "digit is the same as the units digit"
> "digit of the sum of the squares of"
> "is the sum of the lengths of these"
> ```
>
>
> In conclusion, both our method's theoretical design and this direct decontamination analysis confirm that our synthesis process effectively avoids significant contamination from the benchmark test sets. We will add this detailed analysis to the revised manuscript to make this explicit.
>
>
>
> ## Incomplete cost comparison
>
> Thank you for your valuable suggestion regarding the cost comparison in Table 1.  A direct cost comparison with a few methods using open-source models is indeed challenging. Unlike proprietary models, where costs can be directly calculated from API pricing, the cost of running open-source models depends heavily on implementation details, hardware, and specific synthesis settings, making a direct, apples-to-apples comparison difficult.
>
> However, to address your concern, we incorporate a **rough cost comparison** in our analysis. We have selected the **MAmmoTH2** method as a representative open-source baseline, as it synthesizes data by filtering and rewriting pre-training corpora using an open-source model. To quantitatively compare the cost-effectiveness of the two methods, we establish the following assumptions for cost comparison:
>
> - **Base Cost Unit:** The cost of processing *n* tokens with a 7B model is defined as 1 unit.
> - **Task Length:** The average processing length for a standard question-answering task is *n*, while for a raw document processing task, it is *5n*.
> - **Model Cost Multipliers:** 7B=1.0x, Mixtral-8x22B=5.5x(for activated parameters), 32B=4.5x, 72B=10.0x.
>
> **GRIP Method Cost Analysis**:
>
> - Question Generation (32B model): 3M tasks × 4.5x = 13.5M units
> - Solution Generation (7B & 72B models): (2M × 1.0x) + (1M × 10.0x) = 12.0M units
> - Filtering (3x7B models, 2 stages): 6M tasks × 3 × 1.0x = 18.0M units
> - **Total**: ~43.5M units for 2.1M items -> **20.7 units/item**
>
> **MAmmoTH2 Method Cost Analysis**:
>
> - Document Processing (72B model): 18M tasks × 5x length × 10.0x = 900.0M units
> - Data Augmentation (Mixtral 8x22B & 72B): (5M × 5.5x) + (5M × 10.0x) = 77.5M units
> - **Total**: ~977.5M units for 10M items -> **97.75 units/item**
>
> Therefore,  the per-sample synthesis cost of MAmmoTH2 is **4.7 times** higher than that of GRIP. It is crucial to note that this comparison only covers the open-source computation. MAmmoTH2’s pipeline additionally incurs substantial costs from proprietary APIs (GPT-4 and GPT-3.5) for its initial filtering, a cost that GRIP entirely avoids. Therefore, the total cost-effectiveness gap is even larger than our conservative 4.7x estimate. We will add this detailed comparison to the revised manuscript to fully address your concern.
>
> ## Paper Formatting Concerns:
> Thank you for your careful review and for pointing out the formatting error on Line 130. This was indeed a LaTeX label rendering issue where "Appendix ??" was displayed incorrectly. We have now corrected this cross-reference, and it correctly points to Appendix D in the revised manuscript. We appreciate your valuable feedback.

---

> ### Comment · Reviewer_3rLC · 2025-08-04
>
> Thank you for the detailed and thoughtful clarifications. Your additional analyses convincingly address raised concerns.

---

> > ### Author Response · Authors · 2025-08-04
> > **To Reviewer 3rLC**
> >
> > Thank you very much for your positive feedback. Your review comments are very helpful in improving our work, and we are pleased that our additional analyses and clarifications ultimately earned your approval. Thank you again for your valuable time and review.

---

### Official Review · Reviewer_Fk5a · 2025-07-04

**Clarity:** 2
**Significance:** 3
**Originality:** 3
**Rating:** 3
**Confidence:** 3

**Summary:**

This paper presents GRIP, a graph-based framework that synthesizes high-quality mathematical reasoning data by extracting key concepts from seed problems and leveraging both explicit and implicit concept relationships. From 7.5K seed problems, GRIP generates 2.1M diverse question-answer pairs at 99% lower cost than GPT-4 methods while achieving 71.8% novelty rate versus under 40% for existing approaches. Models trained on GRIP-MATH significantly outperform base models and previous synthesis methods on mathematical reasoning benchmarks, demonstrating the effectiveness of concept-driven data generation.

**Questions:**

See Weaknesses.

**Ethical Concerns:**

["NO or VERY MINOR ethics concerns only"]

**Limitations:**

Yes

**Paper Formatting Concerns:**

Line 130, Appendix ??.

**Quality:**

2

**Strengths And Weaknesses:**

## Strengths

1. The motivation is interesting, and the paper is easy to understand.

2. Compared to other data synthesis methods, GRIP used less money to synthesize more data and achieve better results.

## Weaknesses

1. How do you validate the accuracy of the synthesized 2.1M questions?

2. The comparison in Table 2 seems unfair. To properly validate a `training set`’s merit, a reasonable, fair setup should randomly sample the "same number" of Q-A pairs from different corpus, then apply SFT on the same LLM backbone. Table 2's evaluation appears unfair - why do other data synthesis methods experiment on Mistral and Qwen1.5 with fewer training tokens than GRIP?

3. How do you store the constructed KB? Can you provide a visualization interface for all the knowledge comcepts extracted by GRIP and their relationships, using online KG like neo4j?

4. What advantages does using open-source models for data synthesis have over GPT-series. Can you provide performance evidence? I don't think cost is the only reason, since other data synthesis methods using current powerful open-source models might achieve better results than GRIP, so this needs proof.

---

> ### Author Rebuttal · Authors · 2025-07-31
>
> ## On validating data accuracy
>
> Thank you for raising this critical question. Ensuring the accuracy of a synthetically generated dataset of such a large scale (2.1 million) is paramount, and we designed our methodology with this challenge at its core.. Given that manual verification of the entire dataset is infeasible, we designed and implemented a rigorous, multi-stage automated quality control process to maximize the quality of the synthetic data, as detailed in Section 3.4 of our paper. This process is built on three main layers of validation:
>
> - **Evaluation and Filtering of Questions:** We employed a multi-model supervisory framework composed of three advanced open-source mathematical models. Each model scored every generated question based on two dimensions: logical soundness and expressive completeness. Only questions surpassing a high weighted-average score of 0.85 are retained, ensuring the clarity and validity of the problems themselves.
>
> - **Evaluation and Filtering of Solutions:** For the evaluation of solutions, we adopted a more stringent "unanimous consent" mechanism. A solution was accepted **only if** it was judged to be completely correct by all three evaluation models simultaneously. This high-standard design aims to maximize the accuracy of the problem-solving process and the final answers in our dataset.
>
> - **Validation of the Automated Evaluation Framework's Efficacy:** To demonstrate the reliability of the aforementioned automated evaluation process, we manually annotated 500 synthetic samples in our ablation study (Section 4.5, Table 5). The results showed that the judgment accuracy of our three-model framework reached **95.7%This result not only confirms the reliability of our process but also shows it outperforms the powerful, closed-source GPT-4.1 (94.3%) on the same task. This comparative experiment provides strong evidence that our quality control process is both reliable and efficient, even surpassing the accuracy of a costly single closed-source model.
>
> In summary, through the separate evaluation of questions and solutions, strict filtering criteria (weighted scoring and unanimous consent), and validation via manual sampling, we have constructed a closed-loop quality assurance system to ensure the accuracy of the dataset to the greatest extent possible. We will further emphasize the details of this process in the revised manuscript to address your concerns more clearly.
>
> ## On the fairness of the comparison in Table 2
>
> Thank you for your insightful comment regarding the fairness of the comparison in Table 2. We agree that a controlled comparison is vital for validating a training set's merit.
>
> The effectiveness of a data synthesis method is evaluated on two key dimensions: the **quality** of the generated data and the **scalability** of the synthesis process. A primary research goal in this field is to **generate large-scale, high-quality data from limited seed data at a low cost.** Our initial experiments, following the previous works, were designed to demonstrate this scalability by using our full dataset across multiple base models. To provide a more direct and fair comparison, as you suggested, we have isolated the results for all methods on the same model architecture  (Mistral-7B) and present a more granular analysis below. We have broken it down into two parts:
>
> |        | Datasets             | Seed Dataset Count | Dataset Count | Avg      |
> | :----- | :----------------- | :----------------- | :------------ | :------- |
> | ～100K | MetaMath           | 15K                | 395K          | 40.3     |
> |        | WizardMath         | 15K                | 96K           | 38.0     |
> |        | MAmmoTH            | 220K               | 262K          | 39.0     |
> |        | MathCoder          | 15k                | 80K           | 38.8     |
> |        | **GRIP-MATH-mini** | **7.5k**           | **80k**       | **43.0** |
> | >1M    | MathScale          | 20K                | 2M            | 51.2     |
> |        | MAmmoTH2           | >10M               | 10M           | 52.2     |
> |        | **GRIP-MATH**      | **7.5k**           | **2.1M**      | **57.3** |
>
> - **Performance at a Smaller Scale (~100k)**
>
> To directly address your point about comparing datasets of the "same number," we randomly sampled 80k Q-A pairs from our synthesized data to create **GRIP-math-mini**. This allows for an apples-to-apples comparison with datasets of a similar size, like MathCoder (80k). The results show that even when controlling for data quantity, our synthesis method's superior quality leads to better performance, outperforming existing methods in this category. For instance, **GRIP-math-mini (43.0%)** significantly surpasses **MathCoder (38.8%)**.
>
> - **Performance at a Larger Scale (>1M)**
>
> This comparison highlights the second key metric: scalability. When comparing our full 2.1M dataset against other large-scale methods, GRIP-math demonstrates superior efficiency and quality. For example, compared to MAmmoTH2, which filters 10M samples from the massive Common Crawl, our method achieves a 5.1% higher score (57.3% vs 52.2%) using only 7.5k seed examples. Similarly, our method outperforms MathScale (which uses GPT-3.5) by 6.1%. This proves that our synthesis method not only scales data volume effectively but also ensures superior data quality.
>
> We will update the manuscript to include this detailed table and analysis. This will make the evaluation more transparent, comprehensive, and address your valid concerns about fairness. Thank you again for this valuable suggestion.
>
> ## On Knowledge Base storage and visualization:
>
> Thank you for your question. We store the constructed Knowledge Base in a structured and portable **JSON format**. A sample entry looks like this:
>
> ```
> {
>   "key_concept1": [1, 2, 8, 15],
>   "key_concept2": [2, 8, 22, 31, 57]
>   ...
> }
> ```
>
> This indicates, for example, that key_concept1 appears in problems 1, 2, 8, and 15; key_concept2 appears in problems 2, 8, 22, 31 and 57. If two key concepts co-occur in any problem (e.g., both appear in problems 2 and 8), we add an edge between them. The edge weight is defined as the number of problems where both appear together (here, weight = 2).
>
> We completely agree that visualization is crucial. To best enable this, we chose a foundational JSON format. Its simplicity and universality make the knowledge base maximally portable and accessible. More importantly, by using Python scripts based on the methodologies described in Sections 3.1-3.3 of our paper, this JSON data is programmatically processed into a **weighted undirected graph**. This graph structure explicitly defines the nodes (concepts), the relationships between them, and the weights of these relationships. Because these attributes are well-defined, the entire Knowledge Base can be easily imported into various graph analysis tools or databases, such as Neo4j, for further exploration.
>
> Due to rebuttal guidelines that prohibit image uploads, we are unable to provide visualizations or a web interface. However, **inspired by your valuable feedback**, we will incorporate high-resolution static visualizations of key subgraphs in the final version of our paper. These images will serve to intuitively illustrate the structure of our knowledge base, showcasing elements such as the neighborhoods of specific concepts, multi-hop reasoning paths, and concept clusters.
>
> ## On the advantages of using open-source models:
>
> Thanks for your question. To be clear, the open-source models we use for data synthesis and evaluation are weaker than the GPT series, which is the tool used by the competing methods listed in Table 1. While the significant cost savings are a major benefit (as noted in Table 1), the advantages of using a well-designed open-source framework extend far beyond cost. We have summarized the following key advantages:
>
> **Robust Quality**:  Even if the models used in our data synthesis process are weaker than the GPT series, a carefully designed multi-model open-source framework can match and even surpass the quality of a single, powerful closed-source model. We provide direct performance evidence for this in our ablation study (Section 4.5, Table 5). In a critical comparison, our collaborative evaluation framework, which utilizes three open-source models, achieved an accuracy of **95.7%**. This was notably higher than the **94.3%** accuracy achieved when using the powerful, closed-source GPT-4.1 model alone for the same task.
>
> **Stability, Control, and Scalability**: Open-source models provide the stability and control necessary for massive data synthesis projects. Proprietary models like ChatGPT and Gemini have significant usage restrictions, including non-permissive licenses, regional availability limitations, and strict API rate limits. These factors can hinder large-scale generation. Our multi-model collaborative method circumvents these issues, enabling the rapid, stable, and low-cost synthesis of large datasets while avoiding potential licensing and operational risks.
>
> **Flexibility**: The open-source ecosystem is evolving rapidly with the constant emergence of more advanced models (e.g., from DeepSeek, Qwen, Kimi, Llama). Our framework is inherently modular, which grants us the flexibility to easily swap or integrate different state-of-the-art open-source models based on specific task requirements or performance needs.
>
> **Furthermore, data diversity is what truly distinguishes GRIP from other synthesis pipelines.** While other methods might simply rephrase seed data, GRIP's graph-based approach explores **implicit relationships** between knowledge concepts to generate truly novel problems. This is proven by our **71.8% novelty rate (Table 1)** and the **lower seed data similarity visualized in Figure 4**, both confirming that GRIP-MATH is not just a rehash of the original data.
>
> ## Paper Formatting Concerns:
> Thanks! The 'App. ??' error points to Appendix D.

---

> ### Comment · Area_Chair_FDQf · 2025-08-06
>
> Hi Reviewer Fk5A,
>
> Could you please check the author's response and see if it has addressed your questions or concerns? Please kindly point out any remaining issue. It'd be highly appreciated if you could check out other reviews and responses as well.
>
> Thanks, --AC

---

### Note · Authors · 2025-08-12

Dear Area Chair and Reviewers,

We sincerely thank all reviewers for their thorough evaluation and constructive feedback. We are greatly encouraged that reviewers recognized our work as a "novel and significant contribution" with a "very clever idea", noting its "ability to generate diverse problems while avoiding the direct use of seed questions and the use of implicit concept connections makes the approach valuable". The recognition of our "comprehensive experiments" and the value of our large-scale "GRIP-MATH" dataset affirms our core contributions.

In response to the valuable feedback, we conducted additional experiments and analyses to address all raised concerns. Our primary responses include:

(1) Clarifying data accuracy and the effectiveness of our multi-model evaluation. We leveraged new analyses and the ablation study in Table 5 to demonstrate our multi-model evaluation framework's 95.7% accuracy (surpassing GPT-4.1 ), confirming the high quality of the resulting synthetic data.
(2) Highlighting GRIP's effectiveness through robust comparisons. We conducted new comparisons using a size-controlled dataset (GRIP-MATH-mini) for a direct, apples-to-apples evaluation, and extended our testing to more challenging benchmarks like AIME 2024.
(3) Validation of the GRIP framework. A validation experiment showed that synthesized problems strongly adhere to their guiding concepts (88.65% full-match, 98.49% partial-match), while n-gram analysis shows low overlap with existing data (10-gram overlap: 0.63%).
(4) Providing crucial methodological clarifications. We formalized our graph construction algorithm with pseudocode, elaborated on the handling of multi-hop relationships, discussed GRIP's generalization potential, and detailed the multi-faceted advantages of our open-source framework beyond just cost.

We are grateful to all reviewers for their valuable comments, especially the final feedback from three reviewers confirming our rebuttals were 'convincing' and successfully addressed their concerns. We commit to incorporating all discussed suggestions and analyses into the final camera-ready manuscript.

Ultimately, we sincerely hope that our work on the GRIP framework, along with the public release of the GRIP-MATH dataset, will inspire further research in scalable, concept-driven data synthesis and serve as a valuable resource for the entire community. Thank you again for your time and consideration.

---

### Decision · Program_Chairs · 2025-09-17

**Decision:**

Accept (poster)

**Comment:**

Summary:

This paper focuses on data synthesis methods to advance the reasoning capabilities of LLMs. It presents GRIP that synthesizes high-quality and diverse reasoning instructions. GRIP constructs a knowledge graph by extracting high-level concepts from seed data, and leverages both explicit and implicit relationships within the graph to drive large-scale and diverse instruction data synthesis, and employs open-source multi-model supervision to ensure data quality. Compared with existing synthetic data methods, GRIP is shown to have achieved greater scalability and diversity and also significantly reduces costs.  On a variety of mathematical reasoning benchmarks, models trained with the synthesized GRIP-MATH demonstrate substantial improvements over their base models and outperform previous data synthesis methods.

Strengths:

1. The paper is well-written with a clear structure and motivation, making it easy to understand.

2. The proposed method is novel and generates diverse problems without the direct use of seed questions or with a less severe risk of data contamination.

3. The paper conducts extensive experiments on math reasoning to show the effectiveness of the approach and the benefit of the curated dataset.

Weaknesses:

Before the rebuttal, the reviewers raised many clarification questions, most of which have been addressed. I  put a lower weight on the review from Fk5a, due to its lower confidence. In addition, the authors conducted more experiments and analyses during the rebuttal to address concerns related to evaluation on more challenging benchmarks, decontamination, and cost analysis. All reviewers (except Fk5a) are satisfied with the author responses and lean towards accepting the paper (borderline accept - accept).

Additional comments:

1. The authors should make changes in the revised version to reflect the discussions in the rebuttal period. Please add the clarifications, discussions on limitations and future work, as well as additional results/analysis in the revision.

2. Based on the new results on the more challenging math benchmarks (e.g., AIME, provided in the response to reviewer 3rLC), it seems that the method is getting less effective on those benchmarks when using stronger base models such as Qwen2. Could you please comment on that in the revised version?

3. Please kindly release the constructed datasets and open-source the data synthesis approach to benefit the community, if not yet.

Overall, I think this paper makes a valuable contribution and would recommend an acceptance.